# ENSEMW2S: CAN AN ENSEMBLE OF LLMS BE LEVERAGED TO OBTAIN A STRONGER LLM?

**Aakriti Agrawal**[†]     **Mucong Ding**[†]     **Zora Che**[†]     **Chenghao Deng**[†]

**Anirudh Satheesh**[†]     **John Langford**[*]     **Furong Huang**[†]

## ABSTRACT

How can we harness the collective capabilities of multiple Large Language Models (LLMs) to create an even more powerful model? This question forms the foundation of our research, where we propose an innovative approach to weak-to-strong (w2s) generalization—a critical problem in AI alignment. Our work introduces an easy-to-hard (e2h) framework for studying the feasibility of w2s generalization, where weak models trained on simpler tasks collaboratively supervise stronger models on more complex tasks. This setup mirrors real-world challenges, where direct human supervision is limited. To achieve this, we develop a novel AdaBoost-inspired ensemble method, demonstrating that an ensemble of weak supervisors can enhance the performance of stronger LLMs across classification and generative tasks on difficult QA datasets. In several cases, our ensemble approach matches the performance of models trained on ground-truth data, establishing a new benchmark for w2s generalization. We observe an improvement of up to 14% over existing baselines and average improvements of 5% and 4% for binary classification and generative tasks, respectively. This research points to a promising direction for enhancing AI through collective supervision, especially in scenarios where labeled data is sparse or insufficient.

## 1 INTRODUCTION

As AI models, particularly Large Language Models (LLMs), continue to surpass human performance in various domains, a pressing challenge arises: how do we effectively supervise models that exceed our capabilities? This problem, known as super-alignment, is exacerbated by the scarcity of high-quality labeled data, which limits direct human oversight. The key question driving our work is whether weak models, trained on simpler tasks, can be leveraged to instruct and improve stronger models in complex settings—a problem known as weak-to-strong (w2s) generalization.

The concept of w2s generalization was introduced by Burns et al. (2023), where weak models are used to align stronger models in the absence of sufficient ground-truth supervision. However, while this work laid the groundwork, it left several critical challenges unresolved. **(C1) Single Weak Supervisor Limitation.** Prior studies (Burns et al., 2023; Ji et al., 2024; Charikar et al., 2024; Lang et al., 2024) tend to rely on a single weak supervisor, limiting the diversity and robustness of the supervision. A single model's perspective often falls short when attempting to instruct stronger models in more complex tasks, highlighting the need for a more diversified supervisory approach. **(C2) Lack of Focus on Weak Model Enhancement.** Another limitation is that previous research (Burns et al., 2023; Ji et al., 2024; Charikar et al., 2024; Lang et al., 2024) has focused predominantly on improving knowledge transfer from weak to strong models without addressing how to enhance the weak models themselves. This oversight leaves weak models under-optimized, thereby restricting their utility in complex problem settings. **(C3) Overlooking Task Complexity.** Furthermore, while task complexity plays a crucial role in determining how well weak models can supervise stronger ones, most prior work (Sun et al., 2024) has not adequately addressed this issue. For instance, Burns et al. (2023) briefly explored the impact of task complexity using chess data, but a more structured and systematic approach is needed to differentiate between easy and hard tasks and study their effects on supervision.

---

[*]Microsoft

[†]University of Maryland; e-mail: {agrawal5, furongh}@umd.edu

To address these challenges, we propose a novel ensemble-based method designed to improve w2s generalization. Central to our approach is an easy-to-hard (e2h) framework, which extends w2s generalization by focusing on the progression from simpler tasks (easy) to more complex tasks (hard). This mirrors practical scenarios, where human oversight is more feasible for simpler tasks, and weak models must step in to guide stronger models in tackling harder tasks. In this setting, weak models trained on easy data supervise stronger models working on more difficult problems, creating a more pragmatic approach to w2s generalization.

To further enhance the capabilities of weak models, we develop a novel AdaBoost-inspired ensemble method for generation tasks, in addition to classification tasks. By combining the supervision of multiple weak models, we create a more robust and effective supervisory system for stronger LLMs. This ensemble approach overcomes the limitations of single-supervisor systems and introduces a mechanism to refine the weak models themselves, ensuring they can provide meaningful guidance even in complex tasks. Our experiments demonstrate that this ensemble method not only improves the weak models' generalization capabilities but also enables stronger models to achieve performance on par with oracle models trained on high-quality data.

The **main contributions** of this paper are the following:
**(1) We introduce an ensemble method inspired by AdaBoost,** combining weak LLMs to provide stronger supervision for training stronger models. Our approach is validated through experiments on binary classification tasks, where we observe improvements of up to 14% over baselines and an average improvement of 7% across all model pairs, showcasing the feasibility of w2s generalization.
**(2) We extend this framework to supervised fine-tuning tasks for autoregressive LLMs,** where our novel algorithm combines weak LLMs via a voting mechanism that adjusts token probabilities. In several cases, we observe our strong model trained using weak labels to outperform the strong model trained on ground truth, thus enabling effective supervision, even on complex tasks.
**(3) We propose a practical easy-to-hard (e2h) framework for w2s generalization,** where models trained on easy data provide supervision for harder tasks. This setup emphasizes the importance of task complexity and demonstrates significant improvements when weak models guide strong LLMs. For our EnsemW2S-AdaBoost method, along with observing w2s-trained student models outperforming the strong student oracle in several e2h generalization scenarios, we also observe accuracy improvements of up to 10% over baselines and an average improvement of 3.34% and 4.4% for Quartz and ARC data respectively.

## 2   WEAK-TO-STRONG GENERALIZATION VIA EASY-TO-HARD FRAMEWORK

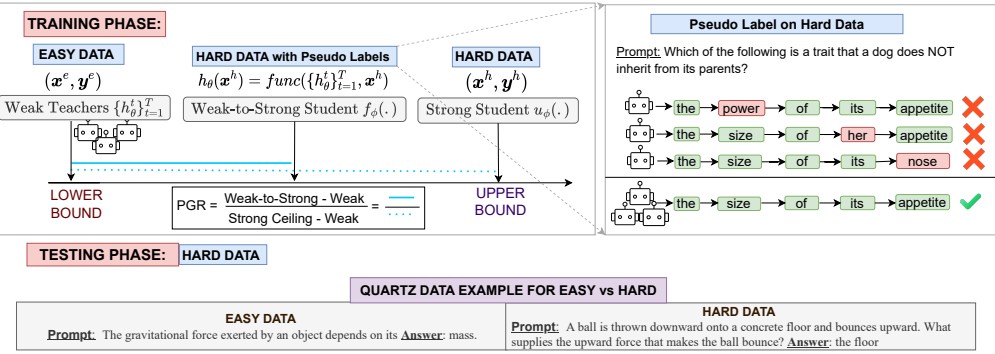

Figure 1: This figure illustrates the complete pipeline of our EnsemW2S method for easy-to-hard generalization using w2s generalization. In a realistic scenario, weak experts are adept at answering easy questions but must supervise strong models to tackle hard problems. **In the leftmost portion**, we show that we train weak models on easy data, strong models on hard data, and transfer models on pseudo labels generated by the weak model on hard data. Ultimately, we aim to increase the Performance Gap Recovered (PGR). **On the right**, we depict how our EnsemW2S-AdaBoost algorithm chooses the correct answer at the token level. **At the bottom**, we provide an example of easy and hard data for the Quartz dataset for e2h generalization, highlighting the importance of distinguishing between easy and hard data for realistic w2s generation.

**The Overall Idea.** We investigate the easy-to-hard framework as a more pragmatic setting to study the (im)possibility of w2s generalization. In this framework, weak models train on simpler tasks and subsequently instruct strong models to tackle more complex challenges, closely mirroring real-world conditions with limited human oversight. Figure 1 explains our idea and pipeline for easy-to-hard generalization using w2s generalization. (Figure 6 in the Appendix provides the detailed algorithmic and data flow). In a realistic scenario, weak experts are proficient in answering easy questions but must supervise strong models to tackle hard problems. We train weak models on easy data and strong models on hard data. A transfer model is trained using pseudo labels generated by the weak model on the hard data. Ultimately, we aim to improve the Performance Gap Recovered (PGR).

## 2.1 THE EASY-TO-HARD FRAMEWORK

**Weak Model $h_\theta$ as the Teacher.** A state-of-the-art LLM $h_\theta$ is trained on a set of 'easy data' that we currently have access to labels, i.e., $(\boldsymbol{x}^e, \boldsymbol{y}^e)$. For example, this could be Go games, math problems, or common sense reasoning questions that we have solutions for. This 'weak teacher' is trained on the labeled easy data $(\boldsymbol{x}^e, \boldsymbol{y}^e)$. Although we refer to this model as a "weak teacher", it is only relatively weak compared to the strong model we aim to obtain. Moreover, the "easy data" is only relatively easy compared to the hard data for which we currently lack solutions. Thus, the easy data may not be simple but slightly easier than the hard data, which are currently unsolvable using existing models.

**Strong Model $u_\phi$ as the Upper Bound.** As an important part of our thought experiment, we establish an upper bound, which is not attainable in practice. Specifically, we assume access to the ground-truth labels of the hard data $(\boldsymbol{x}^h, \boldsymbol{y}^h)$, which is impractical but establishes an upper bound for this thought experiment. A model $u_\phi$, larger than the weak teacher $h_\theta$, is trained on the labeled hard data $(\boldsymbol{x}^h, \boldsymbol{y}^h)$. The reason why $u_\phi$ is larger than $h_\theta$ is that we believe a model strong enough to solve hard questions that no existing models can solve will require high capacity.

**Weak-to-Strong Model $f_\phi$ Obtained in Practice.** To test the weak-to-strong generalization, we will train a weak-to-strong transfer model $f_\phi$ that has the same capacity as the strong model, i.e., the same model size as $u_\phi$, but is not trained under the unrealistic assumption of oracle access to hard labels. Rather, it is trained using weak teacher's feedback. Specifically, we consider using the pseudo-labeled $(\boldsymbol{x}^h, h_\theta(\boldsymbol{x}^h))$ as training data for training the weak-to-strong transfer model $f_\phi$.

## 2.2 EASY AND HARD DATA

**Dataset and Setup.** We use the SciQ dataset (Welbl et al., 2017) for the binary classification task. It is a multiple-choice science question-answer dataset and is also used as one of the NLP classification datasets by Burns et al. (2023). We convert it into binary labels following (Burns et al., 2023). For the supervised fine-tuning (SFT) task on the Q/A dataset, we use ARC (Clark et al., 2018) and Quartz (Tafjord et al., 2019) datasets, which are also multiple-choice question-answer datasets, allowing us to generate multiple-choice pseudo labels. Ding et al. (2024) provide difficulty levels for some common mathematics and programming problems, chess puzzles, and reasoning question datasets, which can be further utilized to expand this work. For details on how we conduct **easy** $(\boldsymbol{x}^e, \boldsymbol{y}^e)$ **and hard** $(\boldsymbol{x}^h, \boldsymbol{y}^h)$ **data split**, refer Appendix Section E.

## 2.3 AN ENSEMBLE OF TEACHERS

In a practical situation, we may face a dearth of strong supervisors but have an abundance of weak supervisors. Previous works (Burns et al., 2023; Ji et al., 2024) have used only one weak supervisor. Our work aims to combine the power of multiple weak supervisors to provide stronger supervision for better weak-to-strong (w2s) generalization. However, combining multiple weak supervisors to improve w2s generalization is challenging. In the following section, we detail how to combine a collection of weak teachers with diverse skill sets to obtain a competitive w2s model that is better than the weak model and ideally reaches or even surpasses the strong model, i.e., the upper bound of performance.

## 3 W2S GENERALIZATION VIA ADABOOST OF DIVERSE TEACHER LLMS

In this section, we introduce our method to boost experts for two tasks: a binary classification task for an NLP dataset and a supervised fine-tuning task for multiple-choice Q/A datasets. A list of important notions is mentioned in Appendix D for reference.

## 3.1 ADABOOST OF WEAK LLM TEACHERS FOR CLASSIFICATION TASKS

This simple thought experiment tests w2s generalization and is the first task tested by Burns et al. (2023). We utilize the vanilla AdaBoost algorithm (Algorithm 2 detailed in the Appendix) to generate answers to a hard question $\boldsymbol{x}^h$ from each weak LLM teacher, i.e., generate $h_\theta^t(\boldsymbol{x}^h)$ for

$t \in \{1, \ldots, T\}$. A weighted "majority vote/aggregation" is implemented to generate a consensus as the answer $\mathbb{1}(\sum_{t=1}^{T} \alpha_t h_\theta^t(\boldsymbol{x}^h) > 0) \in \{0, 1\}$, also known as the pseudo-label, to the hard question $\boldsymbol{x}^h$. Here, the coefficients $\{\alpha_t \mid t \in \{1, \ldots, T\}\}$ are hyperparameters learned during the AdaBoost training. (More details in Appendix Sec F.)

## 3.2 IMPROVING ADABOOST FOR COMPLEX GENERATION TASKS

**Challenges of Applying AdaBoost.** The canonical AdaBoost algorithm assumes a sophisticated ensemble of feedback in the form of scores. However, LLMs are generative AI models known for their remarkable ability to generate coherent, free-form text. Applying the vanilla AdaBoost algorithm directly to generation tasks is challenging because (1) the output is not just a single class label but a sequence of text with no fixed length, and (2) different teachers may generate answers in various formats, making it non-trivial to combine their responses.

**EnsemW2S-AdaBoost: Our modified AdaBoost Algorithm for Multiple-Choice Q/A Task.** To address these challenges, we propose a modified multi-class AdaBoost algorithm where the number of classes corresponds to the vocabulary size. We treat each token as an independent sample, as shown in Algorithm 1, and apply multi-class AdaBoost (Hastie et al., 2009) with modifications, calling our algorithm EnsemW2S-AdaBoost.

*Token-Level Weighting.* The first modification involves generating weights for each token within a sentence sample. We define the initial token-sample weights vector $D_1(i, j) \leftarrow \frac{1}{n}$ for all $i \in [m], j \in [k_i]$, where $n = \sum_{i=1}^{m} k_i$, $k_i$ is the number of tokens in the answer part of each sample $i$, $m$ is the total number of training data samples and $j$ is the $j^{\text{th}}$ token in a particular chosen $i^{\text{th}}$ sample. We update these weights, $D_t(i, j)$, for each iteration $t$ of EnsemW2S-AdaBoost.

*Token-Level Data Sampling.* We sample $S' = \{(\boldsymbol{x}_i'^e, \boldsymbol{y}_i'^e)\}_{i=1}^{m}$ from $S$ using token-sample weights $D_t(i, j)$. By sampling with respect to probability masses $D_t(i, j)$ with repetition, we obtain a set of $n = \sum_{i=1}^{m} k_i$ tokens to train on. However, treating these $n$ sampled tokens as independent training samples is very inefficient. Instead, we "assemble" the sampled tokens back into the sentences they belong to and implement label masks to only train on the sampled tokens in each sentence. Following this method, we can train on sampled tokens with minimal overheads.

*Training and Generating New Weak Teachers.* For each iteration, $t$, of EnsemW2S-AdaBoost we train a new weak teacher model $h_\theta^t$ on the sampled data, $S'$.

*Incorporating Prior Term.* Following Hastie et al. (2009), multi-class boosting uses an additional $\log(c - 1)$ term, where $c$ is the number of classes, in the calculation of the AdaBoost parameter $\alpha$. This term serves two purposes: (1) It enables the generation of weak models with accuracy between 50% and random $\frac{1}{c}$ %, which is crucial for smaller models and challenging tasks that cannot achieve 50% accuracy. (2) It ensures that $\alpha$ remains positive. Bayesian inference is used to provide proof of the benefits of this prior term. Given the large vocabulary size in our case, we introduce a prior term $\log(\frac{1}{1-\epsilon_{pre}} - 1)$, where $\epsilon_{pre}$ is the pre-trained model error of the chosen LLM. This term is sensible because it represents the error before fine-tuning the LLM, effectively replacing the random error baseline. Thus, the final $\alpha$ equation is: $\alpha_t \leftarrow \log(\frac{1-\epsilon_t}{\epsilon_t}) + \log(\frac{1}{1-\epsilon_{pre}} - 1)$.

*Weighted Error Calculation.* Our weighted error equation $\epsilon_t$ also undergoes minor changes. The strict condition for each round of AdaBoost training is that the weighted model error (calculated by comparing each token of each sample) must be less than the pre-training error, i.e., $\epsilon_t < \epsilon_{pre}$. The weighted model error $\epsilon_t$ is defined as, $\epsilon_t = \sum_{i=1}^{m} \sum_{j=1}^{k_i} \mathbb{1}\{h_\theta^t(\boldsymbol{x}_i^e, \boldsymbol{y}_i^{e,j-1}) \neq \boldsymbol{y}_i^{e,j}\} D_t(i, j) < \epsilon_{pre}$. Here, $\boldsymbol{y}_i^{e,j-1}$ is the $(j-1)^{\text{th}}$ ground-truth token in the answer part. The model $h_\theta^t(\boldsymbol{x}_i^e, \boldsymbol{y}_i^{e,j-1})$ predicts the next token and compares it with the ground-truth token $\boldsymbol{y}_i^j$.

*Weight Update Equation.* Our weight update equation for each token is $D_{t+1}(i, j) \leftarrow \frac{1}{Z_t} D_t(i, j) e^{\alpha_t \mathbb{1}\{h_\theta^t(\boldsymbol{x}_i^e, \boldsymbol{y}_i^{e,j-1}) \neq \boldsymbol{y}_i^{e,j}\}}$ where $Z_t$ is a normalization factor calculated by taking the norm of the updated weight vector to ensure $\sum_{i=1}^{m} \sum_{j=1}^{k_i} D_{t+1}(i, j) = 1$.

**Combining Experts to Generate Pseudo Answers for Hard Questions:** To combine the outputs of different experts trained during the various EnsemW2S-AdaBoost rounds, we scale the probability distribution for each token generated by the model $h_\theta^t$ in round $t$ by its corresponding weight $\alpha_t$. Specifically, we multiply $\alpha_t$ by the probability distribution vector of each token. We then aggregate

these weighted distributions across all rounds, normalizing the resulting vector to form a new probability distribution for each token.

---

**Algorithm 1 Main Algorithm: EnsemW2S-AdaBoost**

---

**Input:** An "easy" Q/A training dataset with $m$ examples: $S^e = \{(\boldsymbol{x}_i^e, \boldsymbol{y}_i^e)\}_{i=1}^m$; a pre-trained weak teacher model $h_\theta^0$ parameterized by $\theta$; total number of EnsemW2S-AdaBoost iterations $T$; a "hard" unlabeled (questions only) dataset with $O$ examples: $S^h = \{\boldsymbol{x}_o^h\}_{o=1}^O$

**Output:** Weak-to-Strong Student Model $f_\phi(\cdot)$

1: Initialize Token-Sample Weights: $D_1(i,j) \leftarrow \frac{1}{n}$ for all $i \in [m], j \in [k_i]$, where $k_i$ is the token length in the $i^{\text{th}}$ easy example (i.e., $\boldsymbol{y}_i^e = (\boldsymbol{y}_i^{e,1}, \boldsymbol{y}_i^{e,2}...\boldsymbol{y}_i^{e,k_i})$) and $n = \sum_{i=1}^m k_i$

2: Calculate pre-training error of $h_\theta^0$: $\epsilon_{pre} \leftarrow \sum_{i=1}^m \sum_{j=1}^{k_i} \mathbb{1}\{h_\theta^0(\boldsymbol{x}_i^e, \boldsymbol{y}_i^{e,j-1}) \neq \boldsymbol{y}_i^{e,j}\} D_1(i,j)$

3: **for** $t \leftarrow 1$ to $T$ **do**

4:   Sample $S' = \{(\boldsymbol{x}_i'^e, \boldsymbol{y}_i'^e)\}_{i=1}^m$ from $S$ using token-sample weights $D_t(i,j)$

5:   **Train** a new weak teacher $h_\theta^t$ on $S'$

6:   Calculate $\epsilon_t = \sum_{i=1}^m \sum_{j=1}^{k_i} \mathbb{1}\{h_\theta^t(\boldsymbol{x}_i^e, \boldsymbol{y}_i^{e,j-1}) \neq \boldsymbol{y}_i^{e,j}\} D_t(i,j)$

7:   **if** $\epsilon_t \geq \epsilon_{pre}$ **then**

8:    **break**

9:   **Calculate** $\alpha_t \leftarrow \log \frac{1-\epsilon_t}{\epsilon_t} + \log(\frac{1}{1-\epsilon_{pre}} - 1)$

10:   Update $D_{t+1}(i,j) \leftarrow \frac{1}{Z_t} D_t(i,j) e^{\alpha_t \mathbb{1}\{h_\theta^t(\boldsymbol{x}_i^e, \boldsymbol{y}_i^{e,j-1}) \neq \boldsymbol{y}_i^{e,j}\}}$ for all $i \in [m], j \in [k_i]$, where $Z_t$ is a normalization factor such that $\sum_{i=1}^m \sum_{j=1}^{k_i} D_{t+1}(i,j) = 1$

11: **for** $o \leftarrow 1$ to $O$ **do**

12:   **for** $j \leftarrow 1$ to $k_o$ **do**

13:    Autoregressively generate the $j^{\text{th}}$ token of the "pseudo-answer" $\widehat{\boldsymbol{y}}_o^{h,j} \sim \Delta^{\text{vocab}}(\sum_{t=1}^T \alpha_t \cdot \text{softmax}(h_\theta^t([\boldsymbol{x}_o^h, \widehat{\boldsymbol{y}}_o^{h,1:j-1}])))$, where $\Delta^{\text{vocab}}$ denotes the simplex on the vocabulary

14: **Train** weak-to-strong student model $f_\phi(\cdot)$ on $\{(\boldsymbol{x}_o^h, \widehat{\boldsymbol{y}}_o^h)\}_{o=1}^O$

---

Using this aggregated distribution, we sample the final predicted token. The process is autoregressive, where the $j^{\text{th}}$ token of the "pseudo-answer" is generated as

$$\widehat{\boldsymbol{y}}_o^{h,j} \sim \Delta^{\text{vocab}} \left( \sum_{t=1}^T \alpha_t \cdot \text{softmax} \left( h_\theta^t \left( [\boldsymbol{x}_o^h, \widehat{\boldsymbol{y}}_o^{h,1:j-1}] \right) \right) \right) \tag{1}$$

where $\Delta^{\text{vocab}}$ represents the simplex over the vocabulary.

By combining the outputs of multiple experts, each trained in different EnsemW2S-AdaBoost rounds, the ensemble approach leverages diverse perspectives from the weak models. Each expert contributes its learned strengths, and through weighted aggregation, we diminish the influence of models that are less confident or less effective on certain tokens. This helps reduce variance in the generation process, ensuring that errors from individual weak models are mitigated. The result is a more robust pseudo-labeling system that is better aligned with the true distribution of the hard data, often yielding a performance improvement over any single weak model.

Unlike classification, where scores are combined over a fixed set of classes, generation tasks involve predicting sequences of tokens, where each prediction affects future ones. This makes combining generation probabilities more complex, as errors in early token predictions can propagate throughout the sequence. Additionally, we are aggregating probability distributions over large vocabularies, which introduces computational overhead and potential numerical instability.

Our method addresses these challenges by using a weighted combination of expert models' token probabilities, ensuring that weaker predictions from individual rounds are minimized. By normalizing the aggregated distribution for each token, we maintain valid probability distributions across the vocabulary, effectively reducing the risk of cascading errors during autoregressive generation. This ensemble approach results in a more stable and accurate generation process, mitigating the issues inherent in sequence modeling.

**Pseudo answer generation on multiple-choice datasets:** On multiple-choice datasets, instead of using generated tokens $\widehat{\boldsymbol{y}}^h$ as pseudo answers, we can select one of the choices in the MCQ dataset

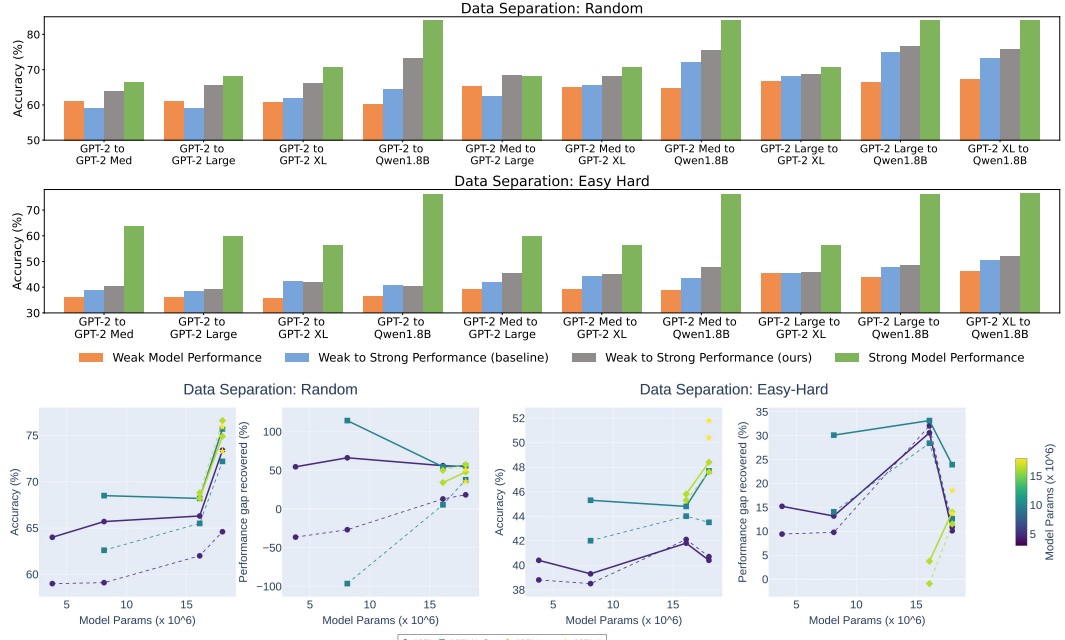

Figure 2: **Binary Classification Task: Top figure** shows a bar plot comparing w2s generalization of our method (grey) with a baseline (blue) from Burns et al. (2023) using accuracy values(%) for different combinations of weak and strong model pairs for random data split (top bar-plot) and easy-hard split(bottom bar-plot). **Bottom figure** shows a line plot comparing the accuracy and performance gap recovered values (PGR). The left two figures are for random data split, while the right two figures are for the easy-hard split to show e2h generalization.

using negative log-likelihood (NLL). Specifically, we calculate the NLL between the choices and $\widehat{\boldsymbol{y}}^h$ and select the choice with the lowest NLL. For datasets without multiple choices, we can directly use $\widehat{\boldsymbol{y}}^h$. For **ablation studies** on our method refer Appendix section G.2

**Train W2S Model:** The strong student model, $f_\phi(\cdot)$, is trained using pseudo answers generated for the hard data $\{(\boldsymbol{x}_o^h, \widehat{\boldsymbol{y}}_o^h)\}_{o=1}^O$. While it might be beneficial to include the labeled easy data in the training process, we adhere to the pipeline established by Burns et al. (2023) by focusing exclusively on the hard examples to maintain consistency.

**Evaluation Metric.** We used two metrics to evaluate this Q/A dataset. One is **(1) Token-wise comparison**, where we compare each predicted token and average the total error, and **(2) Option-wise comparison**, where we compare the negative log-likelihood (NLL) of the correct answer completion with the NLLs of the incorrect answer completions. Accuracy represents the number of entries where the correct answer completion has the lowest NLL among all choices.

## 4 EXPERIMENTAL SETUP

We test two strategies for each task. The first, following Burns et al. (2023), randomly splits the training data into train-weak and train-strong. Train-weak is used to train the weak model. Train-strong is used to train the strong and transfer models using pseudo labels generated using the weak model. The second strategy splits the data into easy (train-weak) and hard (train-strong) subsets, with the same training pipeline, offering a more realistic w2s generalization setup, as discussed in Section 1. Both strategies aim to recover the performance gap (PGR) and maximize the strong model's capability using an ensemble of weak models. The baseline in all experiments uses a single model for w2s generalization, following the principle of Burns et al. (2023). More details in appendix sec G.3.

### 4.1 BINARY CLASSIFICATION TASK

**W2S Results with Random Training Data Splits.** The baseline of this method is a replication of Burns et al. (2023). From Figure 2, by applying AdaBoost, we observe a significant improvement in the weak model accuracy, significantly improving the PGR values. In the case of the GPT-2-medium to GPT-2-large pair, we even see the PGR exceeding 100%, meaning that the transfer model has

outperformed the strong model's performance. This is the ambitious aim of the w2s generalization problem, and our results show that w2s generalization is achievable.

**W2S Results with Easy and Hard Training Data Splits.** From Figure 2, we see that applying AdaBoost significantly improves weak model accuracy, thereby enhancing the PGR values. However, for this holistic e2h generalization problem, we are far from reaching the full capability of a strong model. For very small (GPT-2) and large model pairs (GPT-2-xl and above), we do not see improvement in w2s generalization despite the weak models' accuracy improvements. Overall, we observe an improvement of up to 14% in accuracy compared to the baseline and an average improvement of 6.52% and 3% for random and easy-hard splits, respectively.

**Scaling Law:** In Figure 2 (line plot), we see less PGR recovery for the Qwen-1.8B model even though it is similar in size to GPT-2-xl. Similarly, in the bar plot, we see a drastic difference between the oracle performance of GPT2xl and Qwen-1.8B. This is because the Qwen models series are more capable even after being the same size. Thus, model size is not a good metric, but model capability is a better metric for differentiating between weak and strong models.

**Better metric:** Figure 2 shows the accuracy and PGR plots for both random and easy-hard split. We observe that PGR is not very informative, as it can produce extremely large or even negative values. In the w2s experiments, large values occur because the ensemble of weak models becomes strong enough to match or exceed a strong model, improving w2s generalization. Negative values, seen in baseline experiments, indicate the transfer model performed worse than the weak model, often when the strong model fails to learn and its inductive bias becomes random with pseudo-label training. Similar patterns are seen in Figure 15 and 4. (Refer to Appendix Table 1 and 2 for more details.)

## 4.2 GENERATION TASK FOR MULTIPLE CHOICE DATASET

### 4.2.1 COMPARING WEAK MODEL'S PERFORMANCE

In Figure 3, we compare the performance of a single weak model (dark color) with combined weak models after 5 rounds of EnsemW2S-AdaBoost. Smaller models show greater improvement, which is expected since boosting works best when weak models are diverse. Using EnsemW2S-AdaBoost, smaller models can diversify through the data sampling step; however, larger models tend to learn all possible information and cannot learn something different with each round. We use token-error here since it's a more precise metric to measure improvement in weak models.

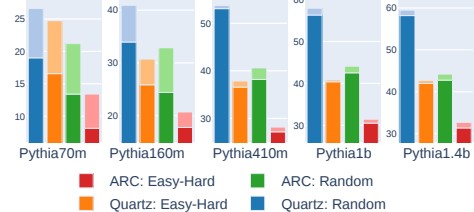

Figure 3: Performance comparison of a single weak model (dark color) with the combined weak models (Lighter hue shows improvement).

### 4.2.2 COMPARING STRONG MODEL'S PERFORMANCE

Here, we use the multiple-choice classification accuracies to calculate the accuracy of all our plots. We show the accuracy values of token-wise metrics in the Appendix tables.

**W2S Results with Random Training Data Splits.** From Figure 4 and 15, we see that w2s training using an ensemble of experts almost consistently outperforms the baseline (single expert). Thus, ensemble learning is beneficial. We can see the trend of accuracy and performance gap recovered for the different model pairs in Figure 4 and 15 for Quartz and ARC datasets, respectively. For Quartz data, we see that our PGR percentage (Figure 4) improves as the model scales up except when the weak model is the smallest sized model (pythia-70m). This could be because the increasing capability difference between the small and large models makes it difficult for the strong model to learn anything from the weak. This trend is the same in the baseline as well as our EnsemW2S. But an important thing to note is that for some cases for both ARC and Quartz data, our method generates a large PGR percentage of >=100%, showing the ability of our w2s method to recover the performance gap.

**W2S Results with Easy-Hard Training Data Splits.** From Figure 4 and 15, we see that w2s training using an ensemble of experts almost consistently outperforms the baseline (single expert). Thus showing that ensemble learning is beneficial. Our method shows more improvement over baseline for easy-hard data split as compared to random split. This is because of two reasons. Firstly, the power of combining weak models using our modified AdaBoost is more useful when all of them are weak

but slightly different from each other. Secondly, by easy and hard splitting, the margin between weak and strong increases more, giving more room for improvement.

We also observe that PGR for e2h generalization is significantly lower, highlighting the complexity of the e2h generalization problem. We hope this work could motivate researchers to build more sophisticated methods for this more complex e2h generalization problem. Another simple observation is as the models become more capable, both the performances (baseline and ours) increase.

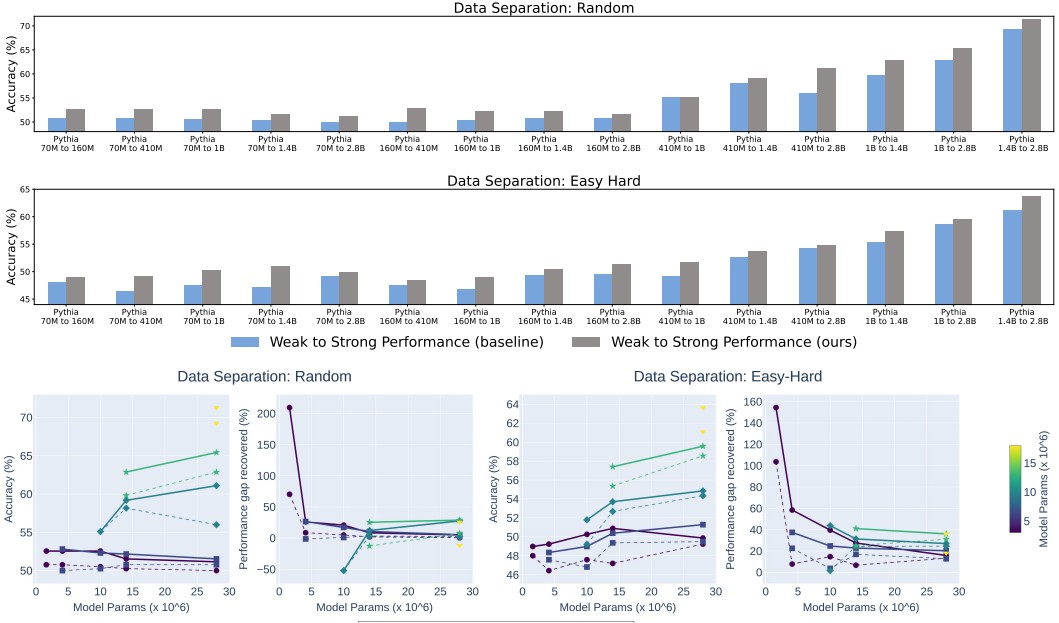

Figure 4: **Generation Task (Quartz Data): Top figure** shows a bar plot comparing the w2s generalization of our method (grey) with a baseline (blue) for various combinations of weak and strong model pairs for the SFT task on Q/A data for random data split (top bar-plot) and easy-hard split (bottom bar-plot). **Bottom figure** shows a line plot comparing accuracy and PGR. The left two figures are for random data split, while the right two are for the easy-hard split to show e2h generalization.

Note: Refer to Appendix Table 3 and 6 for detailed values of our experiments on the Quartz and ARC datasets with random data splits. Bar plots of weak and strong (oracle) model performance for these splits are shown in Appendix Figure 13 and 16. For easy-hard data split, the same details can be found in Appendix Tables 4, 7 and Figure 14 and 17.

### 4.2.3 PERFORMANCE ON HARD DATA AFTER TRAINING ON WEAK VS STRONG DATA

This experiment highlights the importance of e2h with w2s generalization. In Table 5, the Quartz dataset shows significant improvement for larger models when trained on hard data, indicating their better ability to understand complex data. For ARC, all models improve but with a smaller margin, suggesting less disparity between easy and hard samples in the ARC dataset.

| Model Size | Quartz | | ARC | |
|---|---|---|---|---|
| | **Easy Split** | **Hard Split** | **Easy Split** | **Hard Split** |
| pythia-70m | 49.11 | **50.13** | 21.42 | **25.26** |
| pythia-160m | **48.47** | 46.43 | 21.85 | **22.10** |
| pythia-410m | 51.50 | **51.50** | 18.01 | **18.95** |
| pythia-1b | 53.32 | **56.77** | 19.80 | **22.10** |
| pythia-1.4b | 60.34 | **63.78** | 21.42 | **21.42** |
| pythia-2.8b | 66.84 | **70.41** | 25.09 | **26.71** |

Figure 5: Accuracy (%) values for LLMs trained on easy vs hard data and evaluated on hard data.

## 5 CONCLUSION

This paper aims to stimulate discussion on the more holistic problem of w2s generalization by emphasizing e2h generalization. We develop a new AdaBoost-inspired algorithm and conduct a thought experiment on how to combine the "wisdom of the crowd" to improve w2s generalization. We are first to focus on the idea of making the weaks less weak using an ensemble, and test our method for complex SFT tasks. Our method in some cases recovers full strong model capability.

## ACKNOWLEDGEMENTS

Agrawal, Ding, Che, Deng, Satheesh, Langford and Huang are supported by DARPA Transfer from Imprecise and Abstract Models to Autonomous Technologies (TIAMAT) 80321, National Science Foundation NSF-IIS-2147276 FAI, DOD-ONR-Office of Naval Research under award number N00014-22-1-2335, DOD-AFOSR-Air Force Office of Scientific Research under award number FA9550-23-1-0048, DOD-DARPA-Defense Advanced Research Projects Agency Guaranteeing AI Robustness against Deception (GARD) HR00112020007, Adobe, Capital One and JP Morgan faculty fellowships.

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

## A   RELATED WORKS

**Weak-to-Strong**   (Burns et al., 2023) was the first to introduce the problem of weak-to-strong generalization for the super-alignment problem, where the ultimate aim is to elicit the full capabilities of the strong model using supervision only from weak models. (Charikar et al., 2024) provides a theoretical framework for the same with insights on how much w2s improvement can occur, though their work is limited to a few layer neural networks. Similarly, (Lang et al., 2024) provides bounds on expansion properties using finite data distributions for when w2s generalization will happen, but only for simple binary classification tasks. (Zhang et al., 2024) proves that transcendence (exceeding the capability of the model that generates the training data) is possible for low-temperature sampling. Although this setting is not exactly w2s, it sheds light on this direction.

Several works have attempted to solve w2s generalization in LLMs. (Sang et al., 2024) tries to improve this supervision using ensemble learning and scalable oversight for binary classification NLP tasks but cannot observe significant improvement. (Ji et al., 2024) introduces a model that enhances the alignment of LLMs with human intentions by correcting the residual differences between aligned and unaligned answers by training on a query-answer correction dataset. This method boosts w2s generalization using supervisory signal from smaller models to improve the performance of complex systems. In (Sun et al., 2024), the authors propose a scalable approach for e2h generalization which involves training reward models on easier tasks and using them to evaluate performance on harder tasks. (Liu & Alahi, 2024) introduces a method similar to the classical hierarchical mixture of experts, where multiple specialized weak supervisors are used for weak-to-strong generalization instead of a single generalist model. (Bansal et al., 2024) compares large LLM training from data generated using weak (cheap) vs strong (expensive) model in a compute matching way and finds larger data from weaker model to provide better w2s.

Guo et al. (2024) introduces an dynamic adjustable loss function for weak-to-strong supervision. Hase et al. (2024) demonstrates that current language models can achieve high performance on difficult tasks by training on simpler, cleanly labeled data, thus avoiding the high costs and noise associated with hard data labeling. None of these works focused on making the weak teachers, less weak but only focus on improving transfer learning and correction of weak labels. Thus, our method can be combined with all ideas focused on improving transfer learning.

**Ensemble Learning**   Binary Classification Boosting (Freund & Schapire, 1997) and multi-classfication boosting (Hastie et al., 2009) are common ensemble learning algorithms. In (Verga et al., 2024), they use a voting mechanism to combine multiple small LLMs instead of a single large LLM to evaluate another LLM and show it performs better than large LLMs. An extended related work section is present in Appendix A.

**Multi-LLM learning:**   There are numerous works involving the collaboration of multiple LLMs. Chang et al. (2023) proposes Reinforcement Learning with Guided Feedback (RLGF), where a dynamic black-box guide like GPT-3 is used to fine-tune large language models. Rosset et al. (2024) introduces Direct Nash Optimization (DNO), a scalable algorithm that combines contrastive learning with general preference optimization. Cai et al. (2024) presents MEDUSA, an innovative framework designed to accelerate inference in large language models by introducing multiple decoding heads, enabling simultaneous prediction of several tokens, and enhancing efficiency through reduced decoding steps and parallel processing capabilities. Shen et al. (2024) proposes Co-LLM, a collaborative decoding framework that interleaves token-level generations from multiple models. This method optimizes the latent variable model for marginal likelihood, allowing a base model to decide when to generate tokens itself or utilize an assistant model, thereby improving performance across various specialized tasks without direct supervision. Jin et al. (2024) introduces a novel collaborative decoding framework aimed at improving the factuality of large language models by employing a critical token classifier. This approach strategically uses both pre-trained and aligned models to selectively generate critical tokens, significantly enhancing the model's ability to maintain factual accuracy without compromising the diversity of the generated content.

Additionally, Mudgal et al. (2023) introduces Controlled Decoding (CD), a method for aligning language model outputs with desired outcomes using a separate prefix scorer module. This approach allows multi-objective RL without additional training and performs well on benchmarks, bridging the gap between token-level control and sequence-level best-of sampling strategies.

## B    LIMITATION AND FUTURE WORK

(Continue from main manuscript)

*Limitation and Future Work:* This work only explores the supervised fine-tuning phase. While SFT is an important part of the LLM learning pipeline, our future work will focus on developing weak supervision in the reward modeling phase. Another interesting future direction would be to improve the combination of tokens in the decoding phase by replacing the classical AdaBoost algorithm with more adaptive ensemble learning methods. We hope this work sparks discussion on combining multiple LLMs to improve weak-to-strong generalization.

*Computational Overhead:* For fully generative tasks, multiple forward passes are required in an autoregressive manner. At each step, the final voted token is input to all LLMs to predict the next token. This increases generation time, which can be mitigated using efficient decoding algorithms like speculative decoding. Addressing this also forms part of our future work. *Smaller Models:* Another limitation is of all w2s work is they attempt to mimic the weak and strong setting as an analogy to the realistic problem and cannot test on a real human with super-human model.

## C    DETAILED FLOWCHART

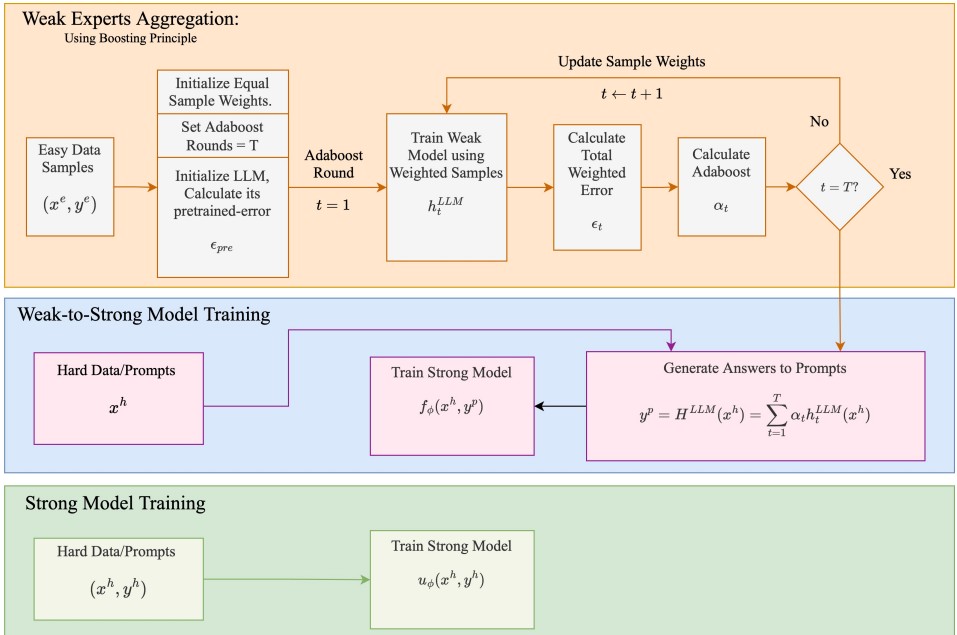

Figure 6: This figure explains our pipeline for easy-to-hard generalization using w2s generalization in complete detail including the algorithm and data flow. We train weak models on easy data and strong models on hard data. A transfer model is trained using pseudo labels generated by the weak model on the hard data. Ultimately, we aim to improve the Performance Gap Recovered (PGR).

## D    IMPORTANT NOTATIONS

Easy Data: $\{(\boldsymbol{x}_i^e, \boldsymbol{y}_i^e)\}_{i=1}^m$

Hard Data: $\{(\boldsymbol{x}_o^h, \boldsymbol{y}_o^h)\}_{o=1}^O$

Total number of Easy Data points: $m$

Total number of Hard Data points: $O$

Total EnsemW2S-AdaBoost Rounds: $T$ Weak Teachers: $\{h_\theta^t\}_{t=1}^T$

Strong Student (Oracle): $u_\phi$

Weak-to-Strong model: $f_\phi$

Total number of tokens in the answer part of each sample $i$: $k_i$

AdaBoost voting parameter: $\{\alpha_t\}_{t=1}^T$

EnsemW2S-AdaBoost token-sample weights for $i^{th}$ sample and $j^{th}$ token: $\{D_t(i,j)\}_{t=1}^T$

Pre-trained Model error: $\epsilon_{pre}$

EnsemW2S-AdaBoost's weighted model error for round $t$: $\epsilon_t$

## E  EASY AND HARD DATA SPLIT.

**Easy** $(x^e, y^e)$ **and Hard** $(x^h, y^h)$ **Data Split:** To generate difficulty ratings for our datasets, we employ the $n$-fold cross-validation method. We train the model on the $(n-1)$ out of $n$ splits of the data and test on the remaining split. We repeat the process $n$ times with different splits for testing each time and aggregate the errors. We use this error value for each sample as its difficulty rating. We split the low difficulty-rated data for weak model training and use the high difficulty-rated data to generate strong model training data and testing data randomly. We follow the same cross-validation method, with different training protocols, for generating difficulty for both binary classification and generation tasks. More details and our difficulty rating plots can be seen in Figures 7, 8, and 9 in the Appendix.

## F  BINARY CLASSIFICATION TASK

AdaBoost utilizes the wisdom of the crowd to obtain a stronger learner. Inspired by its philosophy, we use an ensemble of weak LLM teachers as the "weak learners" to obtain a "stronger learner", i.e., a strong model that improves binary classification tasks, thus achieving weak-to-strong generalization.

**Results and Observations.** As shown in the weak-model performance columns in Table 1 and 2 in the Appendix, the combined weak experts ($T = 2, 3, 4, 5$) demonstrate higher performance than a single weak expert (baseline).

**Training Methodology.** In our method, the weak experts are trained to minimize the error on the reweighted training examples, as detailed in Line 5 of Algorithm 2. The only requirement is that they perform better than random, thus satisfying the well-known weak learning condition. These weak experts represent a practical scenario where, although individually weak, they possess complementary knowledge. Thus, when combined, they have the potential to form a stronger expert.

---

**Algorithm 2** AdaBoost Freund & Schapire (1997)

---

**Input:** Training Dataset $S = \{(x_i, y_i)\}_{i=1}^m \sim D^m$
    $T$ = AdaBoost iterations
    $\vec{D}_1(i) \leftarrow \frac{1}{m} \forall i \in [m]$
    **for** $t \leftarrow 1$ to $T$ **do**
        $h_t$ such that $\epsilon_t = \sum_{i=0}^m \mathbb{1}\{h_t(x_i) \neq y_i\}\vec{D}_t(i) < \frac{1}{2}$
        $\alpha_t \leftarrow \frac{1}{2}\log\frac{1-\epsilon_t}{\epsilon_t}$
        $Z_t \leftarrow 2\sqrt{\epsilon_t(1-\epsilon_t)}$
        $\vec{D}_{t+1} \leftarrow \frac{1}{Z_t}\vec{D}_t e^{-\alpha_t y_i h_t(x_i)}$
        $g \leftarrow \sum_{t=1}^T \alpha_t h_t$
    Return $h(x) = \text{sign}(g)$

---

### F.1  DETAILED RESULTS FOR BINARY CLASSIFICATION TASK WITH $\alpha$ AND $Err_t^{Train}$ IN TABLE 1 AND TABLE 2

Table 1: This table shows weak to strong generalization using random data-splits for sciq dataset. We also study the impact of using ensemble learning methods like AdaBoost, which combines weak learners, for weak to strong training. Each model is trained for 3 epochs and uses an optimized learning rate.

| AdaBoost | Weak Model | Strong Model | Weak-to-Strong Model | $\alpha$ | $Err_t^{Train}$ |
|---|---|---|---|---|---|
| Model Name | GPT-2 | GPT-2 Medium | | | |
| Baseline | 0.610 | 0.665 | 0.590 | 0.455 | 0.287 |
| With AdaBoost (T:02) | 0.613 | 0.665 | 0.619 | 0.488 | 0.274 |
| With AdaBoost (T:03) | 0.614 | 0.665 | 0.609 | 0.463 | 0.284 |
| With AdaBoost (T:04) | 0.611 | 0.665 | 0.622 | 0.467 | 0.282 |
| **With AdaBoost (T:05)** | **0.623** | **0.665** | **0.640** | 0.448 | 0.290 |
| With AdaBoost (T:06) | 0.621 | 0.665 | 0.641 | 0.333 | 0.340 |
| With AdaBoost (T:07) | 0.646 | 0.665 | 0.638 | 0.433 | 0.300 |
| With AdaBoost (T:08) | 0.610 | 0.665 | 0.626 | 0.471 | 0.281 |
| With AdaBoost (T:09) | 0.634 | 0.665 | 0.619 | 0.463 | 0.284 |
| With AdaBoost (T:10) | 0.618 | 0.665 | 0.622 | 0.503 | 0.268 |
| Model Name | GPT-2 | GPT-2 Large | | | |
| Baseline | 0.610 | 0.681 | 0.591 | 0.455 | 0.287 |
| **With AdaBoost (T:02)** | **0.613** | **0.681** | **0.657** | 0.488 | 0.274 |
| With AdaBoost (T:03) | 0.614 | 0.681 | 0.620 | 0.463 | 0.284 |
| With AdaBoost (T:04) | 0.611 | 0.681 | 0.629 | 0.467 | 0.282 |
| With AdaBoost (T:05) | 0.623 | 0.681 | 0.656 | 0.448 | 0.290 |
| With AdaBoost (T:06) | 0.621 | 0.681 | 0.650 | 0.333 | 0.340 |
| With AdaBoost (T:07) | 0.646 | 0.681 | 0.654 | 0.433 | 0.300 |
| With AdaBoost (T:08) | 0.610 | 0.681 | 0.633 | 0.471 | 0.281 |
| With AdaBoost (T:09) | 0.634 | 0.681 | 0.648 | 0.463 | 0.284 |
| With AdaBoost (T:10) | 0.618 | 0.681 | 0.652 | 0.503 | 0.268 |
| Model Name | GPT-2 | GPT-2 XL | | | |
| Baseline | 0.607 | 0.707 | 0.620 | 0.455 | 0.287 |
| With AdaBoost (T:02) | 0.613 | 0.707 | 0.654 | 0.488 | 0.274 |
| With AdaBoost (T:03) | 0.614 | 0.707 | 0.628 | 0.463 | 0.284 |
| **With AdaBoost (T:04)** | **0.611** | **0.707** | **0.663** | 0.467 | 0.282 |
| With AdaBoost (T:05) | 0.623 | 0.707 | 0.645 | 0.448 | 0.290 |
| With AdaBoost (T:06) | 0.621 | 0.707 | 0.648 | 0.333 | 0.340 |
| With AdaBoost (T:07) | 0.646 | 0.707 | 0.649 | 0.433 | 0.300 |
| With AdaBoost (T:08) | 0.610 | 0.707 | 0.653 | 0.471 | 0.281 |
| With AdaBoost (T:09) | 0.634 | 0.707 | 0.657 | 0.463 | 0.284 |
| With AdaBoost (T:10) | 0.618 | 0.707 | 0.654 | 0.503 | 0.268 |
| Model Name | GPT-2 | Qwen1.5-1.8B | | | |
| Baseline | 0.602 | 0.842 | 0.646 | 0.445 | 0.291 |
| With AdaBoost (T:02) | 0.599 | 0.842 | 0.683 | 0.500 | 0.269 |
| With AdaBoost (T:03) | 0.626 | 0.842 | 0.702 | 0.444 | 0.292 |
| With AdaBoost (T:04) | 0.611 | 0.842 | 0.723 | 0.400 | 0.310 |
| With AdaBoost (T:05) | 0.613 | 0.842 | 0.704 | 0.461 | 0.285 |
| **With AdaBoost (T:06)** | **0.613** | **0.842** | **0.734** | 0.417 | 0.303 |
| With AdaBoost (T:07) | 0.603 | 0.842 | 0.712 | 0.422 | 0.301 |
| With AdaBoost (T:08) | 0.608 | 0.842 | 0.717 | 0.319 | 0.346 |
| With AdaBoost (T:09) | 0.614 | 0.842 | 0.712 | 0.405 | 0.308 |
| With AdaBoost (T:10) | 0.606 | 0.842 | 0.712 | 0.360 | 0.328 |
| Model Name | GPT-2 Medium | GPT-2 Large | | | |
| Baseline | 0.653 | 0.681 | 0.626 | 0.705 | 0.196 |
| With AdaBoost (T:02) | 0.656 | 0.681 | 0.643 | 0.624 | 0.223 |
| With AdaBoost (T:03) | 0.646 | 0.681 | 0.639 | 0.674 | 0.206 |
| With AdaBoost (T:04) | 0.663 | 0.681 | 0.664 | 0.645 | 0.216 |
| With AdaBoost (T:05) | 0.645 | 0.681 | 0.654 | 0.690 | 0.201 |
| With AdaBoost (T:06) | 0.652 | 0.681 | 0.667 | 0.619 | 0.225 |
| With AdaBoost (T:07) | 0.650 | 0.681 | 0.665 | 0.722 | 0.191 |
| **With AdaBoost (T:08)** | **0.657** | **0.681** | **0.685** | 0.733 | 0.187 |
| With AdaBoost (T:09) | 0.651 | 0.681 | 0.684 | 0.601 | 0.231 |
| With AdaBoost (T:10) | 0.648 | 0.681 | 0.666 | 0.682 | 0.203 |
| Model Name | GPT-2 Medium | GPT-2 XL | | | |
| Baseline | 0.653 | 0.707 | 0.655 | 0.705 | 0.196 |
| With AdaBoost (T:02) | 0.656 | 0.707 | 0.651 | 0.624 | 0.223 |
| With AdaBoost (T:03) | 0.646 | 0.707 | 0.648 | 0.674 | 0.206 |
| With AdaBoost (T:04) | 0.663 | 0.707 | 0.675 | 0.645 | 0.216 |
| With AdaBoost (T:05) | 0.645 | 0.707 | 0.663 | 0.690 | 0.201 |
| **With AdaBoost (T:06)** | **0.652** | **0.707** | **0.682** | 0.619 | 0.225 |
| With AdaBoost (T:07) | 0.650 | 0.707 | 0.657 | 0.722 | 0.191 |
| With AdaBoost (T:08) | 0.657 | 0.707 | 0.673 | 0.733 | 0.187 |
| With AdaBoost (T:09) | 0.651 | 0.707 | 0.665 | 0.601 | 0.231 |
| With AdaBoost (T:10) | 0.648 | 0.707 | 0.687 | 0.682 | 0.203 |
| Model Name | GPT-2 Medium | Qwen1.5-1.8B | | | |
| Baseline | 0.649 | 0.842 | 0.722 | 0.658 | 0.211 |
| With AdaBoost (T:02) | 0.649 | 0.842 | 0.742 | 0.626 | 0.222 |
| With AdaBoost (T:03) | 0.669 | 0.842 | 0.732 | 0.673 | 0.206 |
| **With AdaBoost (T:04)** | **0.649** | **0.842** | **0.757** | 0.662 | 0.210 |
| With AdaBoost (T:05) | 0.661 | 0.842 | 0.745 | 0.688 | 0.202 |
| With AdaBoost (T:06) | 0.655 | 0.842 | 0.735 | 0.722 | 0.191 |
| With AdaBoost (T:07) | 0.664 | 0.842 | 0.732 | 0.717 | 0.192 |
| With AdaBoost (T:08) | 0.664 | 0.842 | 0.741 | 0.718 | 0.192 |
| With AdaBoost (T:09) | 0.657 | 0.842 | 0.748 | 0.791 | 0.171 |
| With AdaBoost (T:10) | 0.667 | 0.842 | 0.737 | 0.671 | 0.207 |
| Model Name | GPT-2 Large | GPT-2 XL | | | |
| Baseline | 0.673 | 0.707 | 0.682 | 1.675 | 0.034 |
| With AdaBoost (T:02) | 0.658 | 0.707 | 0.675 | 0.974 | 0.125 |
| With AdaBoost (T:03) | 0.671 | 0.707 | 0.687 | 1.091 | 0.101 |
| With AdaBoost (T:04) | 0.671 | 0.707 | 0.684 | 1.080 | 0.103 |
| With AdaBoost (T:05) | 0.668 | 0.707 | 0.687 | 1.033 | 0.112 |
| With AdaBoost (T:06) | 0.675 | 0.707 | 0.683 | 1.133 | 0.094 |
| **With AdaBoost (T:07)** | **0.669** | **0.707** | **0.688** | 1.083 | 0.103 |
| With AdaBoost (T:08) | 0.676 | 0.707 | 0.683 | 1.047 | 0.110 |
| With AdaBoost (T:09) | 0.678 | 0.707 | 0.682 | 1.085 | 0.103 |
| With AdaBoost (T:10) | 0.669 | 0.707 | 0.681 | 1.132 | 0.094 |
| Model Name | GPT-2 Large | Qwen1.5-1.8B | | | |
| Baseline | 0.664 | 0.842 | 0.749 | 1.454 | 0.052 |
| With AdaBoost (T:02) | 0.670 | 0.842 | 0.717 | 0.971 | 0.126 |
| With AdaBoost (T:03) | 0.670 | 0.842 | 0.728 | 0.037 | 0.481 |
| With AdaBoost (T:04) | 0.677 | 0.842 | 0.727 | 1.128 | 0.095 |
| With AdaBoost (T:05) | 0.675 | 0.842 | 0.740 | 1.107 | 0.098 |
| With AdaBoost (T:06) | 0.677 | 0.842 | 0.737 | 0.979 | 0.124 |
| **With AdaBoost (T:07)** | **0.676** | **0.842** | **0.766** | 1.136 | 0.093 |
| With AdaBoost (T:08) | 0.680 | 0.842 | 0.741 | 1.103 | 0.099 |
| With AdaBoost (T:09) | 0.691 | 0.842 | 0.762 | 1.075 | 0.104 |
| With AdaBoost (T:10) | 0.683 | 0.842 | 0.755 | 1.052 | 0.109 |
| Model Name | GPT-2 XL | Qwen1.5-1.8B | | | |
| Baseline | 0.673 | 0.842 | 0.733 | 0.564 | 0.244 |
| With AdaBoost (T:02) | 0.701 | 0.842 | 0.740 | 0.428 | 0.298 |
| With AdaBoost (T:03) | 0.702 | 0.842 | 0.753 | 0.383 | 0.317 |
| With AdaBoost (T:04) | 0.694 | 0.842 | 0.756 | 0.316 | 0.347 |
| **With AdaBoost (T:05)** | **0.704** | **0.842** | **0.759** | 0.260 | 0.373 |
| With AdaBoost (T:06) | 0.693 | 0.842 | 0.757 | 0.288 | 0.360 |
| With AdaBoost (T:07) | 0.708 | 0.842 | 0.755 | 0.277 | 0.365 |
| With AdaBoost (T:08) | 0.706 | 0.842 | 0.761 | 0.223 | 0.391 |
| With AdaBoost (T:09) | 0.700 | 0.842 | 0.748 | 0.252 | 0.377 |
| With AdaBoost (T:10) | 0.703 | 0.842 | 0.747 | 0.258 | 0.374 |

Table 2: This table shows weak to strong generalization using easy and hard data-splits for sciq dataset. We also study the impact of using ensemble learning methods like AdaBoost, which combines weak learners, for weak to strong training. Each model is trained for 3 epochs and uses an optimized learning rate.

| AdaBoost | Weak Model | Strong Model | Weak-to-Strong | $\alpha$ | $Err_t^{Train}$ |
|---|---|---|---|---|---|
| Model Name | GPT-2 | GPT-2 Medium | | | |
| Baseline | 0.362 | 0.638 | 0.388 | 2.178 | 0.013 |
| With AdaBoost (T:02) | 0.356 | 0.638 | 0.382 | 1.790 | 0.027 |
| With AdaBoost (T:03) | 0.343 | 0.638 | 0.386 | 1.953 | 0.020 |
| With AdaBoost (T:04) | 0.361 | 0.638 | 0.385 | 2.014 | 0.018 |
| With AdaBoost (T:05) | 0.361 | 0.638 | 0.382 | 1.534 | 0.044 |
| With AdaBoost (T:06) | 0.365 | 0.638 | 0.393 | 1.588 | 0.040 |
| With AdaBoost (T:07) | 0.365 | 0.638 | 0.402 | 1.474 | 0.050 |
| **With AdaBoost (T:08)** | **0.369** | **0.638** | **0.404** | 1.478 | 0.049 |
| With AdaBoost (T:09) | 0.362 | 0.638 | 0.394 | 1.865 | 0.023 |
| With AdaBoost (T:10) | 0.364 | 0.638 | 0.394 | 1.267 | 0.074 |
| Model Name | GPT-2 | GPT-2 Large | | | |
| Baseline | 0.362 | 0.597 | 0.385 | 2.178 | 0.013 |
| With AdaBoost (T:02) | 0.356 | 0.597 | 0.367 | 1.790 | 0.027 |
| With AdaBoost (T:03) | 0.343 | 0.597 | 0.383 | 1.953 | 0.020 |
| With AdaBoost (T:04) | 0.361 | 0.597 | 0.379 | 2.014 | 0.018 |
| With AdaBoost (T:05) | 0.361 | 0.597 | 0.387 | 1.534 | 0.044 |
| With AdaBoost (T:06) | 0.365 | 0.597 | 0.382 | 1.588 | 0.040 |
| With AdaBoost (T:07) | 0.365 | 0.597 | 0.388 | 1.474 | 0.050 |
| With AdaBoost (T:08) | 0.369 | 0.597 | 0.389 | 1.478 | 0.049 |
| **With AdaBoost (T:09)** | **0.362** | **0.597** | **0.393** | 1.865 | 0.023 |
| With AdaBoost (T:10) | 0.364 | 0.597 | 0.395 | 1.267 | 0.074 |
| Model Name | GPT-2 | GPT-2 XL | | | |
| Baseline | 0.355 | 0.561 | 0.421 | 2.178 | 0.013 |
| With AdaBoost (T:02) | 0.356 | 0.561 | 0.409 | 1.791 | 0.027 |
| With AdaBoost (T:03) | 0.343 | 0.561 | 0.409 | 1.953 | 0.020 |
| With AdaBoost (T:04) | 0.361 | 0.561 | 0.407 | 2.014 | 0.018 |
| **With AdaBoost (T:05)** | **0.361** | **0.561** | **0.418** | 1.534 | 0.044 |
| With AdaBoost (T:06) | 0.365 | 0.561 | 0.409 | 1.588 | 0.040 |
| With AdaBoost (T:07) | 0.365 | 0.561 | 0.407 | 1.474 | 0.050 |
| With AdaBoost (T:08) | 0.369 | 0.561 | 0.413 | 1.478 | 0.049 |
| With AdaBoost (T:09) | 0.362 | 0.561 | 0.410 | 1.865 | 0.023 |
| With AdaBoost (T:10) | 0.364 | 0.561 | 0.409 | 1.267 | 0.074 |
| Model Name | GPT-2 | Qwen1.5-1.8B | | | |
| Baseline | 0.364 | 0.760 | 0.407 | 2.178 | 0.013 |
| With AdaBoost (T:02) | 0.356 | 0.760 | 0.397 | 1.791 | 0.027 |
| With AdaBoost (T:03) | 0.343 | 0.760 | 0.393 | 1.953 | 0.020 |
| With AdaBoost (T:04) | 0.361 | 0.760 | 0.381 | 2.014 | 0.018 |
| With AdaBoost (T:05) | 0.361 | 0.760 | 0.390 | 1.534 | 0.044 |
| With AdaBoost (T:06) | 0.365 | 0.760 | 0.394 | 1.588 | 0.040 |
| With AdaBoost (T:07) | 0.365 | 0.760 | 0.390 | 1.474 | 0.050 |
| With AdaBoost (T:08) | 0.369 | 0.760 | 0.387 | 1.478 | 0.049 |
| With AdaBoost (T:09) | 0.362 | 0.760 | 0.402 | 1.865 | 0.023 |
| **With AdaBoost (T:10)** | **0.364** | **0.760** | **0.404** | 1.267 | 0.074 |
| Model Name | GPT-2 Medium | GPT-2 Large | | | |
| Baseline | 0.391 | 0.597 | 0.420 | 1.511 | 0.046 |
| With AdaBoost (T:02) | 0.448 | 0.597 | 0.438 | 1.571 | 0.041 |
| With AdaBoost (T:03) | 0.426 | 0.597 | 0.405 | 1.483 | 0.049 |
| With AdaBoost (T:04) | 0.454 | 0.597 | 0.437 | 1.601 | 0.039 |
| With AdaBoost (T:05) | 0.448 | 0.597 | 0.428 | 1.334 | 0.065 |
| With AdaBoost (T:06) | 0.465 | 0.597 | 0.444 | 1.249 | 0.076 |
| **With AdaBoost (T:07)** | **0.449** | **0.597** | **0.453** | 1.460 | 0.051 |
| With AdaBoost (T:08) | 0.461 | 0.597 | 0.444 | 1.646 | 0.036 |
| With AdaBoost (T:09) | 0.449 | 0.597 | 0.433 | 1.453 | 0.052 |
| With AdaBoost (T:10) | 0.447 | 0.597 | 0.424 | 1.154 | 0.090 |
| Model Name | GPT-2 Medium | GPT-2 XL | | | |
| Baseline | 0.392 | 0.561 | 0.440 | 1.510 | 0.047 |
| With AdaBoost (T:02) | 0.459 | 0.561 | 0.442 | 1.589 | 0.040 |
| With AdaBoost (T:03) | 0.420 | 0.561 | 0.435 | 1.669 | 0.034 |
| With AdaBoost (T:04) | 0.458 | 0.561 | 0.441 | 1.460 | 0.051 |
| With AdaBoost (T:05) | 0.424 | 0.561 | 0.431 | 1.393 | 0.058 |
| **With AdaBoost (T:06)** | **0.444** | **0.561** | **0.448** | 1.286 | 0.071 |
| With AdaBoost (T:07) | 0.419 | 0.561 | 0.436 | 1.429 | 0.054 |
| With AdaBoost (T:08) | 0.454 | 0.561 | 0.443 | 1.596 | 0.039 |
| With AdaBoost (T:09) | 0.437 | 0.561 | 0.439 | 1.577 | 0.041 |
| With AdaBoost (T:10) | 0.432 | 0.561 | 0.439 | 1.289 | 0.071 |
| Model Name | GPT-2 Medium | Qwen1.5-1.8B | | | |
| Baseline | 0.388 | 0.760 | 0.435 | 1.511 | 0.046 |
| **With AdaBoost (T:02)** | **0.448** | **0.760** | **0.477** | 1.571 | 0.041 |
| With AdaBoost (T:03) | 0.426 | 0.760 | 0.462 | 1.483 | 0.049 |
| With AdaBoost (T:04) | 0.454 | 0.760 | 0.473 | 1.601 | 0.039 |
| With AdaBoost (T:05) | 0.448 | 0.760 | 0.471 | 1.334 | 0.065 |
| With AdaBoost (T:06) | 0.465 | 0.760 | 0.470 | 1.249 | 0.076 |
| With AdaBoost (T:07) | 0.449 | 0.760 | 0.469 | 1.460 | 0.051 |
| With AdaBoost (T:08) | 0.461 | 0.760 | 0.480 | 1.646 | 0.036 |
| With AdaBoost (T:09) | 0.449 | 0.760 | 0.476 | 1.453 | 0.052 |
| With AdaBoost (T:10) | 0.447 | 0.760 | 0.483 | 1.154 | 0.090 |
| Model Name | GPT-2 Large | GPT-2 XL | | | |
| Baseline | 0.454 | 0.561 | 0.453 | 2.981 | 0.003 |
| With AdaBoost (T:02) | 0.451 | 0.561 | 0.455 | 1.791 | 0.027 |
| With AdaBoost (T:03) | 0.458 | 0.561 | 0.451 | 1.954 | 0.020 |
| With AdaBoost (T:04) | 0.463 | 0.561 | 0.447 | 2.220 | 0.012 |
| With AdaBoost (T:05) | 0.471 | 0.561 | 0.452 | 2.145 | 0.014 |
| **With AdaBoost (T:06)** | **0.465** | **0.561** | **0.458** | 1.745 | 0.030 |
| With AdaBoost (T:07) | 0.459 | 0.561 | 0.453 | 1.729 | 0.031 |
| With AdaBoost (T:08) | 0.469 | 0.561 | 0.455 | 1.726 | 0.031 |
| With AdaBoost (T:09) | 0.471 | 0.561 | 0.445 | 1.915 | 0.021 |
| With AdaBoost (T:10) | 0.466 | 0.561 | 0.447 | 2.179 | 0.013 |
| Model Name | GPT-2 Large | Qwen1.5-1.8B | | | |
| Baseline | 0.439 | 0.760 | 0.476 | 2.745 | 0.004 |
| With AdaBoost (T:02) | 0.437 | 0.760 | 0.467 | 1.747 | 0.029 |
| With AdaBoost (T:03) | 0.443 | 0.760 | 0.469 | 1.874 | 0.023 |
| With AdaBoost (T:04) | 0.445 | 0.760 | 0.460 | 2.018 | 0.017 |
| With AdaBoost (T:05) | 0.448 | 0.760 | 0.468 | 2.063 | 0.016 |
| With AdaBoost (T:06) | 0.449 | 0.760 | 0.467 | 1.639 | 0.036 |
| With AdaBoost (T:07) | 0.444 | 0.760 | 0.457 | 1.673 | 0.034 |
| With AdaBoost (T:08) | 0.453 | 0.760 | 0.468 | 1.727 | 0.031 |
| With AdaBoost (T:09) | 0.443 | 0.760 | 0.475 | 2.049 | 0.016 |
| **With AdaBoost (T:10)** | **0.459** | **0.760** | **0.484** | 2.217 | 0.012 |
| Model Name | GPT-2 XL | Qwen1.5-1.8B | | | |
| Baseline | 0.463 | 0.763 | 0.504 | 1.165 | 0.089 |
| With AdaBoost (T:02) | 0.475 | 0.763 | 0.508 | 1.156 | 0.090 |
| With AdaBoost (T:03) | 0.481 | 0.763 | 0.512 | 0.941 | 0.132 |
| With AdaBoost (T:04) | 0.488 | 0.763 | 0.500 | 0.841 | 0.157 |
| **With AdaBoost (T:05)** | **0.481** | **0.763** | **0.518** | 0.821 | 0.162 |
| With AdaBoost (T:06) | 0.494 | 0.763 | 0.514 | 0.776 | 0.175 |
| With AdaBoost (T:07) | 0.483 | 0.763 | 0.499 | 0.801 | 0.168 |
| With AdaBoost (T:08) | 0.489 | 0.763 | 0.513 | 0.687 | 0.202 |
| With AdaBoost (T:09) | 0.492 | 0.763 | 0.516 | 0.832 | 0.159 |
| With AdaBoost (T:10) | 0.481 | 0.763 | 0.519 | 0.636 | 0.219 |

# G GENERATIVE TASK DETAILS

## G.1 DIFFERENT RATING FOR ALL THE DATASETS

We use GPT-2 for binary classification and pythia-160m for SFT task's easy and hard splitting. We use the same training parameters as used in the training of the actual w2s results.

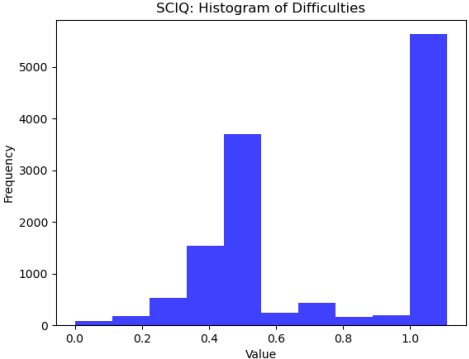

Figure 7: This figure shows the difficulty rating distribution of sciq dataset.

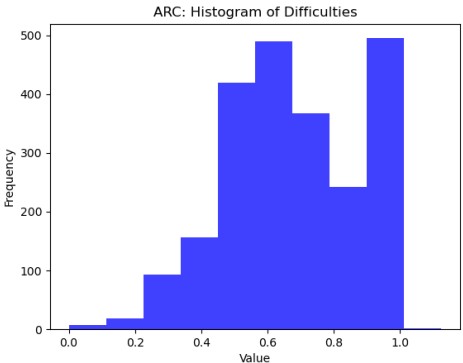

Figure 8: This figure shows difficulty rating distribution of ARC dataset.

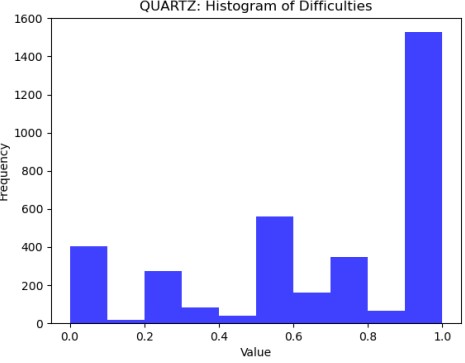

Figure 9: This figure shows difficulty rating distribution of quartz dataset.

## G.2 ABLATION STUDIES FOR GENERATION TASK

**Ablation Studies.** We experimented with combining the logits directly instead of probabilities but did not observe any improvement (refer to Appendix Figure 10). We conducted ablation studies where, instead of treating each token as independent, we used a sliding window of length $L$ while calculating weights and aggregating errors (see Appendix Figure 11 and 12). Different window lengths did not cause significant changes in values, so we ultimately chose a window of $L = 1$. We also explored treating each sample as independent instead of each token as independent in the sample-answer part, finding better results with the latter. This is reasonable since the error calculated using independent-sample weights is less accurate.

### G.2.1 COMPARISON BETWEEN PROBABILITY BASED COMBINATION WITH LOGIT BASED COMBINATION OF THE TOKENS, DURING GENERATION AND EVALUATION OF COMBINED WEAK EXPERTS.

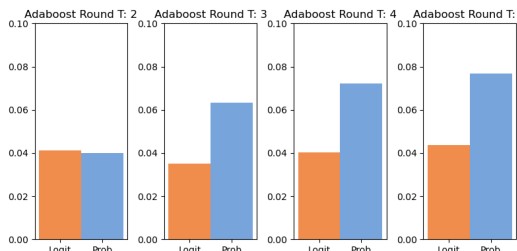

Figure 10: This figure shows a comparison between probability based combination with logit based combination of the tokens.

### G.2.2 COMPARISON BETWEEN DIFFERENT WINDOW LENGTHS FOR "SAMPLE AND TOKEN WEIGHING".

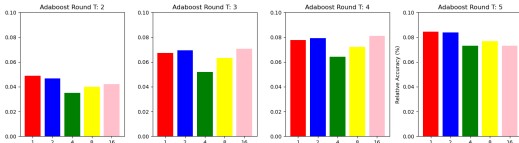

Figure 11: This figure shows a comparison between different token windows for pythia 70m model.

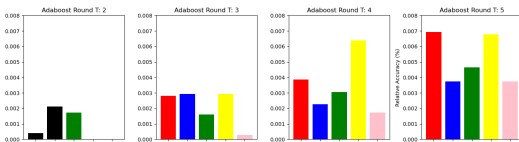

Figure 12: This figure shows a comparison between different token windows for pythia 410m model.

### G.3 EXPERIMENTAL DETAILS

We run AdaBoost/EnsemW2s-AdaBoost 10 times for the binary classification tasks and 5 times for the generation tasks. We pick the best w2s performing round for our plots. However, we observe that all rounds ($n >= 2$) are better than the baseline ($n = 1$). Additionally, we chose single model performance ($n = 1$) for weak model performance.

### G.4 SUPERVISED-FINE TUNING TASK FOR QUARTZ QUESTION-ANSWER DATASET

Table 3: This table shows weak to strong generalization using random data-splits for quartz dataset. We also study the impact of using ensemble learning methods, which combines weak learners, for weak to strong training. Each model is trained for 5 epochs and uses a learning rate of $5x10^{-5}$. The values in this table are generated by aggregating 3 experiments. We show here mean and Standard Error of the Mean values.

| | Weak Model | | | $\alpha$ | Strong Model | | |
|---|---|---|---|---|---|---|---|
| | Token-Avg Acc | Option Acc | Option Acc(on w2s) | | oracle | Token-Avg Acc | Option Acc |
| | Pythia-70m | | | | Pythia-160m | | |
| Baseline | 17.95 ± 0.44 | 50.21 ± 0.23 | 49.7 ± 0.28 | 10.81 ± 0.04 | 50.77 ± 0.26 | 34.3 ± 0.44 | 51.11 ± 0.23 |
| With Adaboost (T:03) | 25.94 ± 0.38 | 50.64 ± 0.39 | 49.43 ± 0.25 | 10.67 ± 0.05 | 50.77 ± 0.26 | 34.17 ± 0.36 | **51.66 ± 0.45** |
| | Pythia-70m | | | | Pythia-410m | | |
| Baseline | 17.95 ± 0.44 | 50.21 ± 0.23 | 49.7 ± 0.28 | 10.81 ± 0.04 | 59.18 ± 0.78 | 50.28 ± 0.44 | 50.68 ± 0.3 |
| With Adaboost (T:04) | 25.22 ± 0.15 | 50.51 ± 0.53 | 49.8 ± 0.14 | 10.68 ± 0.05 | 59.18 ± 0.78 | 50.88 ± 0.18 | **52.42 ± 0.33** |
| | Pythia-70m | | | | Pythia-1b | | |
| Baseline | 17.95 ± 0.44 | 50.21 ± 0.23 | 49.7 ± 0.28 | 10.81 ± 0.04 | 63.35 ± 0.3 | 51.87 ± 0.11 | 50.89 ± 0.16 |
| With Adaboost (T:05) | 26.2 ± 0.06 | 50.55 ± 0.28 | 49.65 ± 0.11 | 10.66 ± 0.04 | 63.35 ± 0.3 | 51.83 ± 0.38 | **51.83 ± 0.31** |
| | Pythia-70m | | | | Pythia-1.4b | | |
| Baseline | 17.89 ± 0.46 | 49.87 ± 0.06 | 49.46 ± 0.35 | 10.82 ± 0.05 | 68.83 ± 1.28 | 51.82 ± 0.05 | 50.17 ± 0.24 |
| With Adaboost (T:04) | 25.32 ± 0.82 | 50.04 ± 0.37 | 49.23 ± 0.27 | 10.7 ± 0.06 | 68.83 ± 1.28 | 51.76 ± 0.17 | **51.45 ± 0.07** |
| | Pythia-70m | | | | Pythia-2.8b | | |
| Baseline | 18.06 ± 0.39 | 49.4 ± 0.39 | 49.73 ± 0.33 | 10.86 ± 0.02 | 73.38 ± 1.02 | 52.28 ± 0.29 | 50.21 ± 0.23 |
| With Adaboost (T:02) | 24.37 ± 0.99 | 50.13 ± 0.4 | 49.48 ± 0.21 | 10.74 ± 0.04 | 73.38 ± 1.02 | 52.3 ± 0.14 | **51.02 ± 0.22** |
| | Pythia-160m | | | | Pythia-410m | | |
| Baseline | 33.51 ± 0.19 | 50.81 ± 1.0 | 49.6 ± 0.27 | 10.03 ± 0.0 | 59.18 ± 0.78 | 50.39 ± 0.3 | 50.68 ± 0.5 |
| With Adaboost (T:04) | 40.85 ± 0.49 | 51.79 ± 0.48 | 49.08 ± 0.32 | 9.81 ± 0.05 | 59.18 ± 0.78 | 50.39 ± 0.18 | **52.13 ± 0.3** |
| | Pythia-160m | | | | Pythia-1b | | |
| Baseline | 33.51 ± 0.19 | 50.81 ± 1.0 | 49.6 ± 0.27 | 10.03 ± 0.0 | 63.35 ± 0.3 | 52.36 ± 0.29 | 50.6 ± 0.33 |
| With Adaboost (T:02) | 40.61 ± 0.8 | 51.36 ± 0.25 | 49.93 ± 0.52 | 9.76 ± 0.05 | 63.35 ± 0.3 | 52.45 ± 0.42 | **51.92 ± 0.31** |
| | Pythia-160m | | | | Pythia-1.4b | | |
| Baseline | 33.42 ± 0.23 | 51.4 ± 0.59 | 49.43 ± 0.41 | 10.03 ± 0.0 | 68.83 ± 1.28 | 52.02 ± 0.2 | 51.02 ± 0.55 |
| With Adaboost (T:03) | 40.87 ± 0.49 | 51.02 ± 0.18 | 49.28 ± 0.13 | 9.75 ± 0.02 | 68.83 ± 1.28 | 52.11 ± 0.39 | **53.02 ± 0.55** |
| | Pythia-160m | | | | Pythia-2.8b | | |
| Baseline | 33.42 ± 0.23 | 51.4 ± 0.59 | 49.43 ± 0.41 | 10.03 ± 0.0 | 73.17 ± 0.88 | 52.82 ± 0.02 | 51.45 ± 0.5 |
| With Adaboost (T:04) | 41.13 ± 0.51 | 51.23 ± 0.4 | 49.65 ± 0.14 | 9.78 ± 0.06 | 73.17 ± 0.88 | 52.51 ± 0.3 | **51.74 ± 0.17** |
| | Pythia-410m | | | | Pythia-1b | | |
| Baseline | 52.71 ± 0.24 | 59.27 ± 0.46 | 55.54 ± 0.49 | 10.0 ± 0.01 | 63.35 ± 0.3 | 53.39 ± 0.2 | 56.21 ± 0.76 |
| With Adaboost (T:02) | 53.39 ± 0.17 | 58.5 ± 0.33 | 55.91 ± 0.35 | 9.69 ± 0.08 | 63.35 ± 0.3 | 53.87 ± 0.46 | **56.42 ± 0.56** |
| | Pythia-410m | | | | Pythia-1.4b | | |
| Baseline | 52.9 ± 0.09 | 59.65 ± 0.15 | 55.66 ± 0.51 | 9.98 ± 0.02 | 68.83 ± 1.28 | 53.33 ± 0.74 | 56.34 ± 0.9 |
| With Adaboost (T:02) | 53.26 ± 0.27 | 58.8 ± 0.42 | 56.11 ± 0.34 | 9.66 ± 0.08 | 68.83 ± 1.28 | 54.14 ± 0.63 | **57.7 ± 0.61** |
| | Pythia-410m | | | | Pythia-2.8b | | |
| Baseline | 52.13 ± 0.64 | 58.29 ± 1.1 | 55.94 ± 0.3 | 9.89 ± 0.06 | 73.38 ± 1.02 | 54.38 ± 0.31 | 55.74 ± 0.73 |
| With Adaboost (T:04) | 53.39 ± 0.19 | 59.18 ± 0.42 | 55.32 ± 0.51 | 9.85 ± 0.05 | 73.38 ± 1.02 | 55.71 ± 0.53 | **59.01 ± 0.94** |
| | Pythia-1b | | | | Pythia-1.4b | | |
| Baseline | 55.65 ± 0.52 | 61.99 ± 0.51 | 58.6 ± 1.13 | 9.85 ± 0.01 | 68.62 ± 0.12 | 55.33 ± 0.31 | 58.93 ± 0.68 |
| With Adaboost (T:03) | 56.81 ± 0.47 | 62.12 ± 0.43 | 58.14 ± 0.85 | 9.74 ± 0.11 | 68.62 ± 0.12 | 55.99 ± 0.16 | **61.69 ± 0.57** |
| | Pythia-1b | | | | Pythia-2.8b | | |
| Baseline | 55.54 ± 0.6 | 62.12 ± 0.51 | 58.55 ± 1.14 | 9.84 ± 0.01 | 73.3 ± 0.3 | 57.26 ± 0.3 | 61.52 ± 1.38 |
| With Adaboost (T:02) | 57.09 ± 0.41 | 62.84 ± 0.12 | 59.0 ± 0.62 | 9.63 ± 0.02 | 73.3 ± 0.3 | 58.1 ± 0.08 | **63.99 ± 0.93** |
| | Pythia-1.4b | | | | Pythia-2.8b | | |
| Baseline | 57.11 ± 0.45 | 69.64 ± 0.97 | 66.87 ± 1.1 | 9.87 ± 0.02 | 73.76 ± 0.67 | 59.34 ± 0.24 | 67.94 ± 0.78 |
| With Adaboost (T:02) | 59.17 ± 0.12 | 70.66 ± 0.06 | 67.29 ± 0.77 | 9.65 ± 0.03 | 73.76 ± 0.67 | 59.3 ± 0.34 | **68.92 ± 1.06** |

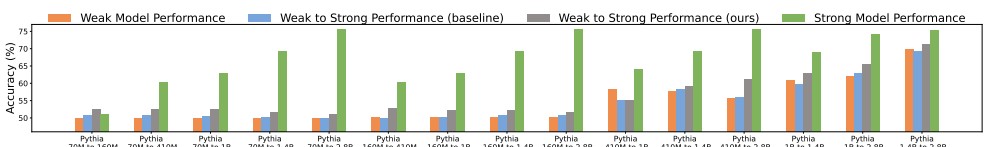

Figure 13: **Quartz Dataset (Random):** This figure shows bar plots comparing accuracy values of weak model performance, w2s model performance (baseline and ours) and strong model performance (oracle) for one specific run of experiments. Values are also mentioned in table 5.

Table 4: This table shows weak to strong generalization using easy-hard data-splits for quartz dataset. We also study the impact of using ensemble learning methods, which combines weak learners, for weak to strong training. Each model is trained for 5 epochs and uses a learning rate of $5 \times 10^{-5}$. The values in this table are generated by aggregating 3 experiments. We show here mean and Standard Error of the Mean values.

| | Weak Model | | | | Strong Model | | |
|---|---|---|---|---|---|---|---|
| | Token-Avg Acc | Option Acc | Option Acc(on w2s) | $\alpha$ | oracle | Token-Avg Acc | Option Acc |
| | Pythia-70m | | | | Pythia-160m | | |
| Baseline | 16.27 ± 0.14 | 48.0 ± 0.51 | 49.21 ± 0.05 | 10.53 ± 0.0 | 47.11 ± 0.28 | 29.24 ± 0.18 | 49.11 ± 0.39 |
| With Adaboost (T:03) | 23.31 ± 0.9 | 47.11 ± 0.31 | 49.23 ± 0.41 | 10.43 ± 0.03 | 47.11 ± 0.28 | 29.24 ± 0.25 | **49.32 ± 0.23** |
| | Pythia-70m | | | | Pythia-410m | | |
| Baseline | 16.27 ± 0.14 | 48.0 ± 0.51 | 49.21 ± 0.05 | 10.53 ± 0.0 | 52.3 ± 0.39 | 43.63 ± 0.29 | 47.32 ± 0.36 |
| With Adaboost (T:04) | 23.81 ± 1.01 | 47.66 ± 0.5 | 49.06 ± 0.2 | 10.42 ± 0.02 | 52.3 ± 0.39 | 43.53 ± 0.44 | **48.13 ± 0.47** |
| | Pythia-70m | | | | Pythia-1b | | |
| Baseline | 16.27 ± 0.14 | 48.0 ± 0.51 | 49.21 ± 0.05 | 10.53 ± 0.0 | 55.91 ± 0.37 | 47.48 ± 0.23 | 47.92 ± 0.23 |
| With Adaboost (T:05) | 24.64 ± 0.22 | 47.49 ± 0.49 | 49.41 ± 0.38 | 10.39 ± 0.0 | 55.91 ± 0.37 | 45.5 ± 0.74 | **49.74 ± 0.24** |
| | Pythia-70m | | | | Pythia-1.4b | | |
| Baseline | 16.07 ± 0.22 | 48.17 ± 0.43 | 49.38 ± 0.14 | 10.58 ± 0.04 | 65.35 ± 0.66 | 46.25 ± 0.61 | 47.96 ± 0.34 |
| With Adaboost (T:04) | 23.79 ± 0.55 | 46.94 ± 0.18 | 49.58 ± 0.27 | 10.44 ± 0.04 | 65.35 ± 0.66 | 45.53 ± 0.2 | **50.68 ± 0.17** |
| | Pythia-70m | | | | Pythia-2.8b | | |
| Baseline | 16.12 ± 0.21 | 48.85 ± 0.48 | 49.75 ± 0.32 | 10.63 ± 0.04 | 70.2 ± 0.17 | 48.08 ± 0.18 | 48.85 ± 0.31 |
| With Adaboost (T:02) | 22.96 ± 0.75 | 47.02 ± 0.12 | 49.36 ± 0.11 | 10.5 ± 0.05 | 70.2 ± 0.17 | 48.58 ± 0.16 | **49.87 ± 0.06** |
| | Pythia-160m | | | | Pythia-410m | | |
| Baseline | 25.61 ± 0.33 | 47.75 ± 0.35 | 49.83 ± 0.29 | 9.96 ± 0.02 | 52.3 ± 0.39 | 42.75 ± 0.91 | 47.75 ± 0.61 |
| With Adaboost (T:04) | 29.63 ± 0.55 | 47.02 ± 0.09 | 48.47 ± 0.3 | 9.7 ± 0.09 | 52.3 ± 0.39 | 43.78 ± 0.14 | **48.42 ± 0.12** |
| | Pythia-160m | | | | Pythia-1b | | |
| Baseline | 25.61 ± 0.33 | 47.75 ± 0.35 | 49.83 ± 0.29 | 9.96 ± 0.02 | 55.91 ± 0.37 | 46.08 ± 0.38 | 49.36 ± 0.53 |
| With Adaboost (T:02) | 28.96 ± 0.23 | 46.43 ± 0.18 | 48.49 ± 0.11 | 9.69 ± 0.09 | 55.91 ± 0.37 | 44.7 ± 0.58 | **49.15 ± 0.73** |
| | Pythia-160m | | | | Pythia-1.4b | | |
| Baseline | 25.76 ± 0.43 | 47.15 ± 0.15 | 49.26 ± 0.2 | 9.96 ± 0.02 | 65.35 ± 0.66 | 45.83 ± 0.64 | 49.7 ± 0.85 |
| With Adaboost (T:03) | 28.83 ± 0.84 | 46.56 ± 0.27 | 48.17 ± 0.14 | 9.64 ± 0.06 | 65.35 ± 0.66 | 45.4 ± 0.44 | **50.0 ± 0.22** |
| | Pythia-160m | | | | Pythia-2.8b | | |
| Baseline | 26.46 ± 0.25 | 47.49 ± 0.33 | 48.98 ± 0.14 | 10.02 ± 0.03 | 70.2 ± 0.17 | 48.03 ± 0.13 | 49.4 ± 0.3 |
| With Adaboost (T:04) | 29.61 ± 0.51 | 46.6 ± 0.25 | 48.69 ± 0.47 | 9.54 ± 0.03 | 70.2 ± 0.17 | 48.4 ± 0.29 | **50.3 ± 0.41** |
| | Pythia-410m | | | | Pythia-1b | | |
| Baseline | 36.73 ± 0.39 | 51.06 ± 0.39 | 53.26 ± 0.38 | 10.07 ± 0.01 | 55.91 ± 0.37 | 46.6 ± 0.38 | 50.72 ± 0.68 |
| With Adaboost (T:02) | 38.11 ± 0.44 | 49.36 ± 0.21 | 51.66 ± 0.35 | 9.76 ± 0.14 | 55.91 ± 0.37 | 46.4 ± 0.35 | **52.09 ± 0.3** |
| | Pythia-410m | | | | Pythia-1.4b | | |
| Baseline | 37.23 ± 0.27 | 51.11 ± 0.4 | 53.19 ± 0.42 | 10.04 ± 0.03 | 65.35 ± 0.66 | 47.73 ± 0.78 | 53.66 ± 0.56 |
| With Adaboost (T:02) | 38.31 ± 0.23 | 50.17 ± 0.44 | 51.56 ± 0.22 | 9.53 ± 0.09 | 65.35 ± 0.66 | 48.35 ± 0.18 | **53.36 ± 0.5** |
| | Pythia-410m | | | | Pythia-2.8b | | |
| Baseline | 37.13 ± 0.23 | 51.02 ± 0.47 | 52.87 ± 0.21 | 10.03 ± 0.03 | 70.2 ± 0.17 | 48.48 ± 0.36 | 54.47 ± 0.16 |
| With Adaboost (T:04) | 38.13 ± 0.26 | 49.87 ± 0.68 | 51.49 ± 0.28 | 9.6 ± 0.04 | 70.2 ± 0.17 | 49.05 ± 0.14 | **55.36 ± 0.47** |
| | Pythia-1b | | | | Pythia-1.4b | | |
| Baseline | 40.3 ± 0.46 | 54.51 ± 0.73 | 54.25 ± 0.26 | 10.33 ± 0.08 | 66.67 ± 0.72 | 47.0 ± 0.22 | 56.76 ± 0.58 |
| With Adaboost (T:03) | 40.75 ± 0.67 | 53.36 ± 0.92 | 53.61 ± 0.44 | 11.0 ± 0.72 | 66.67 ± 0.72 | 47.25 ± 0.32 | **57.23 ± 0.37** |
| | Pythia-1b | | | | Pythia-2.8b | | |
| Baseline | 40.33 ± 0.44 | 54.08 ± 1.07 | 54.33 ± 0.19 | 10.33 ± 0.08 | 73.09 ± 0.42 | 49.2 ± 0.2 | 58.08 ± 0.38 |
| With Adaboost (T:02) | 40.53 ± 0.34 | 52.34 ± 0.09 | 53.39 ± 0.2 | 11.68 ± 0.75 | 73.09 ± 0.42 | 49.48 ± 0.3 | **59.35 ± 0.52** |
| | Pythia-1.4b | | | | Pythia-2.8b | | |
| Baseline | 42.2 ± 1.12 | 59.69 ± 0.83 | 62.39 ± 1.06 | 10.3 ± 0.1 | 73.17 ± 0.38 | 51.22 ± 0.5 | 62.46 ± 0.91 |
| With Adaboost (T:02) | 42.98 ± 0.64 | 59.82 ± 0.51 | 61.38 ± 0.48 | 10.52 ± 0.35 | 73.17 ± 0.38 | 51.72 ± 0.37 | **63.01 ± 0.28** |

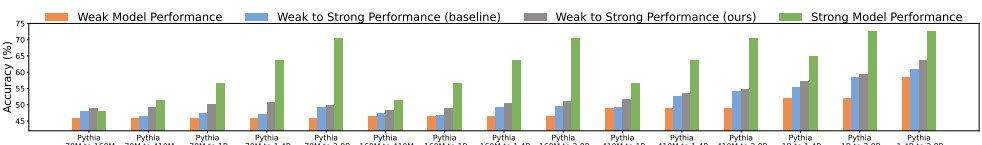

Figure 14: **Quartz Dataset (Easy-Hard):** This figure shows bar plots comparing accuracy values of weak model performance, w2s model performance (baseline and ours) and strong model performance (oracle) for one specific run of experiments. Values are also mentioned in table 5.

Table 5: This table shows weak to strong generalization using random as well as easy-hard data-splits for quartz dataset. As compared to previous tables 3 and 4, here we run experiment once and note the improvement of our method with respect to the baseline.

| Weak Model Size | Strong Model Size | Data Separation: Random | | | | | Data Separation: Easy-Hard | | | | |
| | | Weak Model Performance | Strong Model Performance | W2S Performance | | Improv(%) | Weak Model Performance | Strong Model Performance | W2S Performance | | Improv(%) |
| | | | | Baseline | Ours | | | | Baseline | Ours | |
| 70M | 160M | 0.4987 | 0.5115 | 0.5077 | 0.5255 | 3.5% | 0.4668 | 0.4847 | 0.48 | 0.4898 | 2% |
| 70M | 410M | 0.4987 | 0.602 | 0.5077 | 0.5255 | 3.5% | 0.4885 | 0.4949 | 0.4643 | 0.4923 | 6% |
| 70M | 1B | 0.4987 | 0.6276 | 0.5051 | 0.5255 | 4% | 0.486 | 0.4758 | 0.4758 | 0.5026 | 5.6% |
| 70M | 1.4B | 0.4987 | 0.6926 | 0.5026 | 0.5153 | 2.5% | 0.486 | 0.4974 | 0.4719 | 0.5089 | 7.8% |
| 70M | 2.8B | 0.4987 | 0.7551 | 0.5 | 0.5115 | 2.3% | 0.4872 | 0.5 | 0.4923 | 0.4987 | 1.3% |
| 160M | 410M | 0.5013 | 0.6008 | 0.5 | 0.5281 | 5.6% | 0.4694 | 0.4923 | 0.4758 | 0.4834 | 1.6% |
| 160M | 1B | 0.5013 | 0.6403 | 0.5026 | 0.523 | 4.1% | 0.4872 | 0.5 | 0.4681 | 0.4898 | 4.6% |
| 160M | 1.4B | 0.5013 | 0.713 | 0.5077 | 0.5217 | 2.8% | 0.4668 | 0.5051 | 0.4936 | 0.5038 | 2.1% |
| 160M | 2.8B | 0.5013 | 0.7117 | 0.5077 | 0.5153 | 1.5% | 0.4847 | 0.4847 | 0.4949 | 0.5128 | 3.6% |
| 410M | 1B | 0.5689 | 0.6403 | 0.551 | 0.551 | 0% | 0.4936 | 0.4911 | 0.4921 | 0.5179 | 5.2% |
| 410M | 1.4B | 0.5778 | 0.6926 | 0.5816 | 0.5918 | 1.8% | 0.4936 | 0.4949 | 0.5268 | 0.537 | 1.9% |
| 410M | 2.8B | 0.5561 | 0.7551 | 0.5599 | 0.611 | 9.1% | 0.5013 | 0.5115 | 0.5434 | 0.5485 | 0.9% |
| 1B | 1.4B | 0.6084 | 0.6888 | 0.5982 | 0.6288 | 5.1% | 0.5472 | 0.4923 | 0.5536 | 0.574 | 3.7% |
| 1B | 2.8B | 0.6197 | 0.7398 | 0.6288 | 0.6543 | 4.1% | 0.5497 | 0.5026 | 0.5855 | 0.5957 | 1.7% |
| 1.4B | 2.8B | 0.699 | 0.7538 | 0.6926 | 0.713 | 2.9% | 0.588 | 0.477 | 0.6161 | 0.6288 | 2.1% |

### G.4.1 SUPERVISED-FINE TUNING TASK FOR ARC QUESTION-ANSWER DATASET

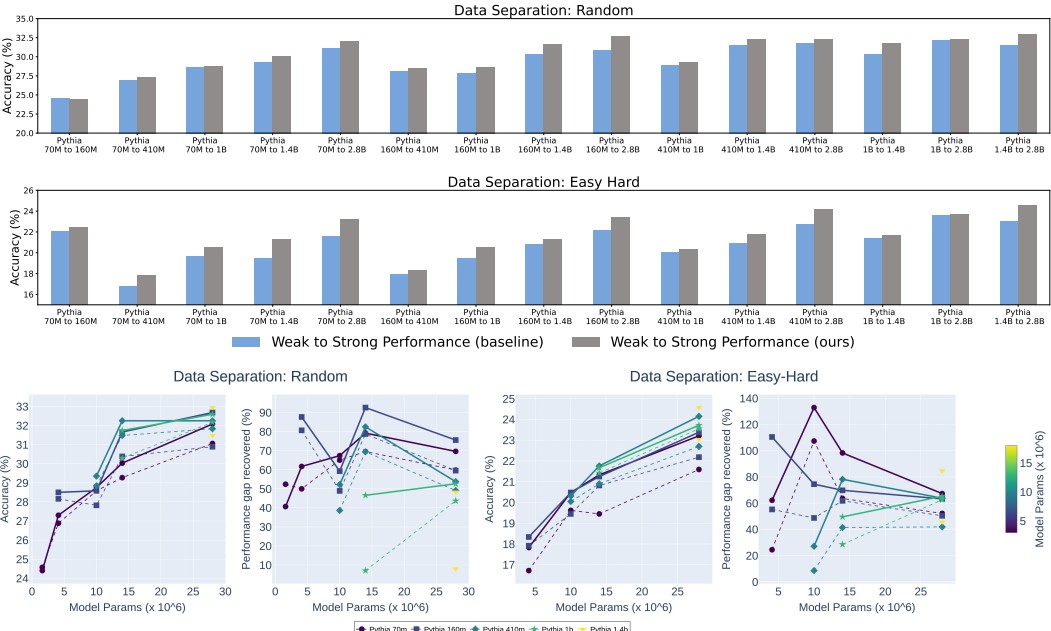

Figure 15: **Generation Task (ARC Data): Top figure** shows a bar plot comparing the w2s generalization of our method (grey) with a baseline (blue) for various combinations of weak and strong model pairs for the SFT task on Q/A data for random data split (top bar-plot) and easy-hard split (bottom bar-plot). **Bottom figure** shows a line plot comparing accuracy and PGR. The left two figures are for random data split, while the right two are for the easy-hard split to show e2h generalization.

Table 6: This table shows weak to strong generalization using random data-splits for arc dataset. We also study the impact of using ensemble learning methods, which combines weak learners, for weak to strong training. Each model is trained for 5 epochs and uses a learning rate of $5x10^{-5}$. The values in this table are generated by aggregating 3 experiments. We show here mean and Standard Error of the Mean values.

| | Weak Model | | | | Strong Model | | |
|---|---|---|---|---|---|---|---|
| | Token-Avg Acc | Option Acc | Option Acc(on w2s) | $\alpha$ | oracle | Token-Avg Acc | Option Acc |
| | Pythia-70m | | | | Pythia-160m | | |
| Baseline | 13.28 ± 0.05 | 25.31 ± 0.1 | 25.76 ± 0.94 | 10.73 ± 0.03 | 24.12 ± 0.48 | 26.91 ± 0.1 | 24.46 ± 0.06 |
| With Adaboost (T:03) | 17.93 ± 0.78 | 24.75 ± 0.76 | 25.82 ± 0.69 | 10.68 ± 0.02 | 24.12 ± 0.48 | 27.15 ± 0.36 | **24.23 ± 0.08** |
| | Pythia-70m | | | | Pythia-410m | | |
| Baseline | 13.28 ± 0.05 | 25.31 ± 0.1 | 25.76 ± 0.94 | 10.73 ± 0.03 | 28.61 ± 0.08 | 41.29 ± 0.1 | 27.25 ± 0.24 |
| With Adaboost (T:04) | 17.94 ± 0.88 | 24.97 ± 0.69 | 25.82 ± 0.69 | 10.67 ± 0.04 | 28.61 ± 0.08 | 41.61 ± 0.02 | **27.27 ± 0.3** |
| | Pythia-70m | | | | Pythia-1b | | |
| Baseline | 13.28 ± 0.05 | 25.31 ± 0.1 | 25.76 ± 0.94 | 10.73 ± 0.03 | 31.11 ± 0.02 | 45.13 ± 0.11 | 28.33 ± 0.18 |
| With Adaboost (T:05) | 19.7 ± 1.18 | 24.92 ± 0.28 | 26.23 ± 0.49 | 10.65 ± 0.04 | 31.11 ± 0.02 | 45.17 ± 0.11 | **28.52 ± 0.09** |
| | Pythia-70m | | | | Pythia-1.4b | | |
| Baseline | 13.35 ± 0.06 | 25.06 ± 0.14 | 24.39 ± 0.42 | 10.77 ± 0.06 | 32.34 ± 0.3 | 45.21 ± 0.24 | 29.86 ± 0.28 |
| With Adaboost (T:04) | 19.75 ± 1.16 | 24.26 ± 0.56 | 25.7 ± 0.65 | 10.68 ± 0.05 | 32.34 ± 0.3 | 45.33 ± 0.14 | **30.35 ± 0.13** |
| | Pythia-70m | | | | Pythia-2.8b | | |
| Baseline | 13.42 ± 0.11 | 24.63 ± 0.13 | 23.97 ± 0.55 | 10.77 ± 0.05 | 35.18 ± 0.02 | 48.07 ± 0.12 | 30.94 ± 0.13 |
| With Adaboost (T:02) | 19.88 ± 0.56 | 24.52 ± 0.49 | 24.87 ± 0.81 | 10.68 ± 0.04 | 35.18 ± 0.02 | 47.75 ± 0.08 | **31.43 ± 0.43** |
| | Pythia-160m | | | | Pythia-410m | | |
| Baseline | 25.5 ± 0.66 | 24.12 ± 0.45 | 26.06 ± 0.68 | 9.89 ± 0.03 | 29.18 ± 0.04 | 41.39 ± 0.14 | 27.5 ± 0.27 |
| With Adaboost (T:04) | 31.95 ± 0.47 | 24.94 ± 0.29 | 25.88 ± 0.64 | 9.74 ± 0.03 | 29.18 ± 0.04 | 41.28 ± 0.03 | **27.7 ± 0.34** |
| | Pythia-160m | | | | Pythia-1b | | |
| Baseline | 25.5 ± 0.66 | 24.12 ± 0.45 | 26.06 ± 0.68 | 9.89 ± 0.03 | 31.26 ± 0.44 | 45.12 ± 0.05 | 28.24 ± 0.18 |
| With Adaboost (T:02) | 32.25 ± 0.21 | 24.52 ± 0.34 | 26.06 ± 0.57 | 9.66 ± 0.01 | 31.26 ± 0.44 | 45.18 ± 0.14 | **28.47 ± 0.24** |
| | Pythia-160m | | | | Pythia-1.4b | | |
| Baseline | 24.74 ± 0.14 | 23.97 ± 0.36 | 25.76 ± 0.51 | 9.86 ± 0.02 | 32.25 ± 0.35 | 45.01 ± 0.1 | 30.55 ± 0.07 |
| With Adaboost (T:03) | 32.55 ± 0.21 | 24.46 ± 0.22 | 26.12 ± 0.8 | 9.66 ± 0.01 | 32.25 ± 0.35 | 45.23 ± 0.05 | **30.86 ± 0.33** |
| | Pythia-160m | | | | Pythia-2.8b | | |
| Baseline | 25.43 ± 0.66 | 24.34 ± 0.09 | 26.0 ± 0.32 | 9.86 ± 0.02 | 35.44 ± 0.06 | 47.88 ± 0.02 | 31.03 ± 0.15 |
| With Adaboost (T:04) | 32.6 ± 0.03 | 24.23 ± 0.18 | 26.47 ± 0.53 | 9.66 ± 0.02 | 35.44 ± 0.06 | 47.77 ± 0.08 | **31.68 ± 0.41** |
| | Pythia-410m | | | | Pythia-1b | | |
| Baseline | 39.76 ± 0.3 | 27.85 ± 0.52 | 24.33 ± 0.97 | 9.39 ± 0.02 | 30.97 ± 0.08 | 44.94 ± 0.08 | 28.9 ± 0.12 |
| With Adaboost (T:02) | 40.69 ± 0.14 | 28.27 ± 0.11 | 24.33 ± 0.59 | 9.01 ± 0.04 | 30.97 ± 0.08 | 44.76 ± 0.14 | **29.41 ± 0.08** |
| | Pythia-410m | | | | Pythia-1.4b | | |
| Baseline | 39.66 ± 0.22 | 27.82 ± 0.53 | 24.09 ± 0.8 | 9.39 ± 0.02 | 32.82 ± 0.27 | 45.54 ± 0.03 | 30.26 ± 0.56 |
| With Adaboost (T:02) | 40.82 ± 0.13 | 28.9 ± 0.21 | 24.51 ± 0.59 | 9.01 ± 0.04 | 32.82 ± 0.27 | 45.66 ± 0.09 | **30.94 ± 0.53** |
| | Pythia-410m | | | | Pythia-2.8b | | |
| Baseline | 39.57 ± 0.24 | 28.01 ± 0.69 | 24.69 ± 0.44 | 9.39 ± 0.01 | 35.86 ± 0.26 | 48.06 ± 0.15 | 31.15 ± 0.3 |
| With Adaboost (T:04) | 40.56 ± 0.11 | 28.7 ± 0.34 | 25.34 ± 1.12 | 9.03 ± 0.07 | 35.86 ± 0.26 | 48.22 ± 0.12 | **31.88 ± 0.27** |
| | Pythia-1b | | | | Pythia-1.4b | | |
| Baseline | 42.31 ± 0.2 | 30.35 ± 0.24 | 28.02 ± 0.76 | 9.53 ± 0.02 | 32.65 ± 0.43 | 45.41 ± 0.06 | 30.26 ± 0.22 |
| With Adaboost (T:03) | 43.22 ± 0.13 | 31.68 ± 0.55 | 27.79 ± 0.71 | 9.37 ± 0.01 | 32.65 ± 0.43 | 45.44 ± 0.06 | **31.28 ± 0.22** |
| | Pythia-1b | | | | Pythia-2.8b | | |
| Baseline | 42.2 ± 0.29 | 30.46 ± 0.16 | 27.73 ± 0.89 | 9.53 ± 0.02 | 35.12 ± 0.26 | 48.12 ± 0.06 | 32.14 ± 0.02 |
| With Adaboost (T:02) | 43.61 ± 0.2 | 31.17 ± 0.93 | 27.79 ± 0.76 | 9.26 ± 0.02 | 35.12 ± 0.26 | 48.2 ± 0.08 | **32.54 ± 0.08** |
| | Pythia-1.4b | | | | Pythia-2.8b | | |
| Baseline | 42.39 ± 0.37 | 33.42 ± 0.37 | 30.65 ± 1.82 | 9.48 ± 0.03 | 35.12 ± 0.26 | 48.35 ± 0.11 | 32.42 ± 0.44 |
| With Adaboost (T:02) | 43.58 ± 0.27 | 33.5 ± 0.22 | 30.71 ± 1.48 | 11.07 ± 0.84 | 35.12 ± 0.26 | 48.29 ± 0.13 | **33.19 ± 0.24** |

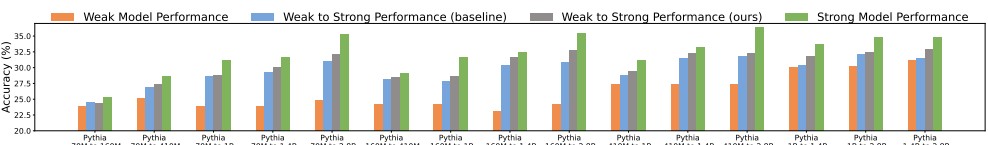

Figure 16: **ARC Dataset (Random):** This figure shows bar plots comparing accuracy values of weak model performance, w2s model performance (baseline and ours) and strong model performance (oracle) for one specific run of experiments. Values are also mentioned in table 8.

Table 7: This table shows weak to strong generalization using easy-hard data-splits for ARC dataset. We also study the impact of using ensemble learning methods, which combines weak learners, for weak to strong training. Each model is trained for 5 epochs and uses a learning rate of $5 \times 10^{-5}$. The values in this table are generated by aggregating 3 experiments. We show here mean and Standard Error of the Mean values.

| | Weak Model | | | $\alpha$ | Strong Model | | |
|---|---|---|---|---|---|---|---|
| | Token-Avg Acc | Option Acc | Option Acc(on w2s) | | oracle | Token-Avg Acc | Option Acc |
| | Pythia-70m | | | | Pythia-160m | | |
| Baseline | $8.17 \pm 0.06$ | $22.5 \pm 0.33$ | $27.85 \pm 0.57$ | $10.45 \pm 0.0$ | $22.3 \pm 0.16$ | $17.88 \pm 0.11$ | $22.27 \pm 0.32$ |
| With Adaboost (T:03) | $13.35 \pm 0.54$ | $22.81 \pm 0.29$ | $27.78 \pm 0.46$ | $10.35 \pm 0.02$ | $22.3 \pm 0.16$ | $17.87 \pm 0.17$ | $\mathbf{22.56 \pm 0.06}$ |
| | Pythia-70m | | | | Pythia-410m | | |
| Baseline | $8.17 \pm 0.06$ | $22.5 \pm 0.33$ | $27.85 \pm 0.57$ | $10.45 \pm 0.0$ | $19.28 \pm 0.15$ | $28.92 \pm 0.14$ | $17.06 \pm 0.31$ |
| With Adaboost (T:04) | $14.53 \pm 0.72$ | $22.93 \pm 0.17$ | $27.96 \pm 0.46$ | $10.32 \pm 0.0$ | $19.28 \pm 0.15$ | $28.84 \pm 0.05$ | $\mathbf{18.0 \pm 0.07}$ |
| | Pythia-70m | | | | Pythia-1b | | |
| Baseline | $8.17 \pm 0.06$ | $22.5 \pm 0.33$ | $27.85 \pm 0.57$ | $10.45 \pm 0.0$ | $21.5 \pm 0.24$ | $32.05 \pm 0.13$ | $19.96 \pm 0.15$ |
| With Adaboost (T:05) | $12.95 \pm 0.88$ | $22.58 \pm 0.38$ | $28.03 \pm 0.21$ | $10.35 \pm 0.02$ | $21.5 \pm 0.24$ | $31.84 \pm 0.08$ | $\mathbf{20.45 \pm 0.06}$ |
| | Pythia-70m | | | | Pythia-1.4b | | |
| Baseline | $8.23 \pm 0.1$ | $22.61 \pm 0.42$ | $27.37 \pm 0.42$ | $10.45 \pm 0.0$ | $21.76 \pm 0.14$ | $32.98 \pm 0.04$ | $20.45 \pm 0.42$ |
| With Adaboost (T:04) | $12.65 \pm 0.05$ | $23.24 \pm 0.06$ | $28.32 \pm 0.76$ | $10.33 \pm 0.01$ | $21.76 \pm 0.14$ | $32.95 \pm 0.17$ | $\mathbf{21.28 \pm 0.02}$ |
| | Pythia-70m | | | | Pythia-2.8b | | |
| Baseline | $8.33 \pm 0.1$ | $23.24 \pm 0.23$ | $27.19 \pm 0.47$ | $10.45 \pm 0.0$ | $26.59 \pm 0.13$ | $35.98 \pm 0.09$ | $22.78 \pm 0.51$ |
| With Adaboost (T:02) | $14.28 \pm 0.15$ | $23.26 \pm 0.22$ | $28.27 \pm 0.14$ | $10.37 \pm 0.01$ | $26.59 \pm 0.13$ | $35.86 \pm 0.28$ | $\mathbf{23.15 \pm 0.2}$ |
| | Pythia-160m | | | | Pythia-410m | | |
| Baseline | $17.46 \pm 0.16$ | $21.73 \pm 0.35$ | $26.95 \pm 0.1$ | $9.61 \pm 0.0$ | $19.11 \pm 0.37$ | $28.8 \pm 0.23$ | $18.15 \pm 0.15$ |
| With Adaboost (T:04) | $20.57 \pm 0.1$ | $22.16 \pm 0.2$ | $27.19 \pm 0.5$ | $9.22 \pm 0.02$ | $19.11 \pm 0.37$ | $28.9 \pm 0.11$ | $\mathbf{18.43 \pm 0.04}$ |
| | Pythia-160m | | | | Pythia-1b | | |
| Baseline | $17.46 \pm 0.16$ | $21.73 \pm 0.35$ | $26.95 \pm 0.1$ | $9.61 \pm 0.0$ | $21.59 \pm 0.07$ | $32.06 \pm 0.06$ | $19.65 \pm 0.1$ |
| With Adaboost (T:02) | $20.47 \pm 0.09$ | $22.27 \pm 0.29$ | $27.31 \pm 0.51$ | $9.24 \pm 0.01$ | $21.59 \pm 0.07$ | $32.07 \pm 0.12$ | $\mathbf{20.17 \pm 0.14}$ |
| | Pythia-160m | | | | Pythia-1.4b | | |
| Baseline | $17.61 \pm 0.07$ | $22.84 \pm 0.58$ | $27.79 \pm 0.64$ | $9.61 \pm 0.0$ | $22.33 \pm 0.34$ | $33.11 \pm 0.1$ | $21.19 \pm 0.15$ |
| With Adaboost (T:03) | $20.31 \pm 0.24$ | $22.5 \pm 0.36$ | $27.79 \pm 0.42$ | $9.27 \pm 0.06$ | $22.33 \pm 0.34$ | $33.01 \pm 0.05$ | $\mathbf{21.25 \pm 0.28}$ |
| | Pythia-160m | | | | Pythia-2.8b | | |
| Baseline | $17.64 \pm 0.06$ | $23.09 \pm 0.54$ | $27.91 \pm 0.59$ | $9.6 \pm 0.01$ | $26.82 \pm 0.1$ | $35.83 \pm 0.36$ | $22.44 \pm 0.11$ |
| With Adaboost (T:04) | $20.3 \pm 0.19$ | $23.01 \pm 0.43$ | $27.73 \pm 0.25$ | $9.26 \pm 0.06$ | $26.82 \pm 0.1$ | $36.06 \pm 0.07$ | $\mathbf{23.35 \pm 0.1}$ |
| | Pythia-410m | | | | Pythia-1b | | |
| Baseline | $27.3 \pm 0.16$ | $18.8 \pm 0.21$ | $31.01 \pm 0.51$ | $9.24 \pm 0.0$ | $21.33 \pm 0.04$ | $32.06 \pm 0.07$ | $20.05 \pm 0.08$ |
| With Adaboost (T:02) | $28.07 \pm 0.12$ | $18.35 \pm 0.21$ | $32.2 \pm 0.31$ | $8.68 \pm 0.09$ | $21.33 \pm 0.04$ | $32.36 \pm 0.05$ | $\mathbf{20.34 \pm 0.06}$ |
| | Pythia-410m | | | | Pythia-1.4b | | |
| Baseline | $27.5 \pm 0.14$ | $18.54 \pm 0.32$ | $31.6 \pm 0.21$ | $9.24 \pm 0.0$ | $22.36 \pm 0.3$ | $33.47 \pm 0.07$ | $21.13 \pm 0.1$ |
| With Adaboost (T:02) | $28.09 \pm 0.08$ | $18.17 \pm 0.28$ | $31.78 \pm 0.4$ | $8.67 \pm 0.09$ | $22.36 \pm 0.3$ | $33.18 \pm 0.11$ | $\mathbf{21.47 \pm 0.12}$ |
| | Pythia-410m | | | | Pythia-2.8b | | |
| Baseline | $27.48 \pm 0.13$ | $18.12 \pm 0.13$ | $31.66 \pm 0.17$ | $9.25 \pm 0.01$ | $26.03 \pm 0.21$ | $36.13 \pm 0.09$ | $23.07 \pm 0.18$ |
| With Adaboost (T:04) | $27.96 \pm 0.11$ | $18.09 \pm 0.2$ | $31.07 \pm 0.27$ | $8.69 \pm 0.08$ | $26.03 \pm 0.21$ | $35.93 \pm 0.09$ | $\mathbf{24.06 \pm 0.15}$ |
| | Pythia-1b | | | | Pythia-1.4b | | |
| Baseline | $30.64 \pm 0.17$ | $21.22 \pm 0.72$ | $32.5 \pm 0.6$ | $9.38 \pm 0.01$ | $22.01 \pm 0.21$ | $33.13 \pm 0.11$ | $21.5 \pm 0.07$ |
| With Adaboost (T:03) | $30.41 \pm 0.42$ | $21.11 \pm 0.22$ | $32.68 \pm 0.56$ | $10.98 \pm 0.78$ | $22.01 \pm 0.21$ | $33.31 \pm 0.03$ | $\mathbf{21.53 \pm 0.08}$ |
| | Pythia-1b | | | | Pythia-2.8b | | |
| Baseline | $30.64 \pm 0.17$ | $21.22 \pm 0.72$ | $32.5 \pm 0.6$ | $9.38 \pm 0.01$ | $25.51 \pm 0.2$ | $36.14 \pm 0.11$ | $23.75 \pm 0.16$ |
| With Adaboost (T:02) | $31.11 \pm 0.12$ | $21.67 \pm 0.18$ | $33.21 \pm 0.56$ | $9.4 \pm 0.24$ | $25.51 \pm 0.2$ | $36.13 \pm 0.13$ | $\mathbf{23.75 \pm 0.06}$ |
| | Pythia-1.4b | | | | Pythia-2.8b | | |
| Baseline | $31.09 \pm 0.12$ | $22.27 \pm 0.55$ | $34.05 \pm 0.1$ | $9.31 \pm 0.01$ | $25.26 \pm 0.11$ | $36.13 \pm 0.05$ | $23.49 \pm 0.2$ |
| With Adaboost (T:02) | $31.56 \pm 0.1$ | $21.79 \pm 0.44$ | $34.35 \pm 0.59$ | $10.89 \pm 0.65$ | $25.26 \pm 0.11$ | $36.36 \pm 0.2$ | $\mathbf{24.37 \pm 0.16}$ |

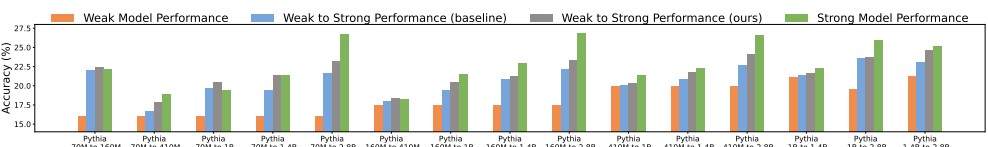

Figure 17: **ARC Dataset (Easy-Hard):** This figure shows bar plots comparing accuracy values of weak model performance, w2s model performance (baseline and ours) and strong model performance (oracle) for one specific run of experiments. Values are also mentioned in table 8.

Table 8: This table shows weak to strong generalization using random as well as easy-hard data-splits for ARC dataset. As compared to previous tables 6 and 7, here we run experiment once and note the improvement of our method with respect to the baseline.

| Weak Model Size (Pythia) | Strong Model Size (Pythia) | Data Separation: Random | | | | Improv (%) | Data Separation: Easy-Hard | | | | Improv (%) |
|---|---|---|---|---|---|---|---|---|---|---|---|
| | | Weak Model Performance | Strong Model Performance | W2S Performance | | | Weak Model Performance | Strong Model Performance | W2S Performance | | |
| | | | | Baseline | Ours | | | | Baseline | Ours | |
| 70M | 160M | 0.2381 | 0.2526 | 0.2457 | 0.244 | -0.7 | 0.16 | 0.221 | 0.2201 | 0.2244 | 2 |
| 70M | 410M | 0.2509 | 0.2867 | 0.2688 | 0.273 | 1.6 | 0.16 | 0.1894 | 0.1672 | 0.1783 | 6.6 |
| 70M | 1B | 0.2381 | 0.3114 | 0.2858 | 0.2875 | 0.6 | 0.16 | 0.1937 | 0.1962 | 0.2048 | 4.4 |
| 70M | 1.4B | 0.2381 | 0.3166 | 0.2927 | 0.3003 | 2.6 | 0.16 | 0.2124 | 0.1945 | 0.2133 | 9.7 |
| 70M | 2.8B | 0.2483 | 0.3524 | 0.3106 | 0.3208 | 3.3 | 0.16 | 0.2671 | 0.2159 | 0.2321 | 7.5 |
| 160M | 410M | 0.2423 | 0.291 | 0.2816 | 0.285 | 1.2 | 0.175 | 0.1826 | 0.1792 | 0.1834 | 2.3 |
| 160M | 1B | 0.2423 | 0.3157 | 0.2782 | 0.2858 | 2.7 | 0.175 | 0.215 | 0.1945 | 0.2048 | 5.3 |
| 160M | 1.4B | 0.2312 | 0.3234 | 0.3038 | 0.3166 | 4.2 | 0.175 | 0.2287 | 0.2082 | 0.2125 | 2.1 |
| 160M | 2.8B | 0.2423 | 0.3541 | 0.3089 | 0.3268 | 5.8 | 0.175 | 0.2679 | 0.2218 | 0.2338 | 5.4 |
| 410M | 1B | 0.2739 | 0.3114 | 0.2884 | 0.2935 | 1.8 | 0.1993 | 0.2133 | 0.2005 | 0.2031 | 1.3 |
| 410M | 1.4B | 0.2739 | 0.3328 | 0.3148 | 0.3225 | 2.4 | 0.1993 | 0.2227 | 0.209 | 0.2176 | 4.1 |
| 410M | 2.8B | 0.2739 | 0.3643 | 0.3183 | 0.3225 | 1.3 | 0.1993 | 0.2654 | 0.227 | 0.2415 | 6.4 |
| 1B | 1.4B | 0.3003 | 0.337 | 0.3029 | 0.3174 | 4.8 | 0.2108 | 0.2227 | 0.2142 | 0.2167 | 1.2 |
| 1B | 2.8B | 0.3012 | 0.3481 | 0.3217 | 0.3259 | 1.3 | 0.1954 | 0.2594 | 0.2355 | 0.2372 | 0.7 |
| 1.4B | 2.8B | 0.3119 | 0.3481 | 0.3148 | 0.3294 | 4.6 | 0.2125 | 0.2517 | 0.2304 | 0.2457 | 6.6 |

## H BROADER IMPACT

The proposed framework for weak-to-strong (w2s) generalization using ensembles of weak language models (LLMs) has significant implications across various domains. By demonstrating that multiple weak supervisors can effectively train more powerful models, our research addresses the critical challenge of superalignment, potentially transforming how advanced AI systems are developed and supervised. This approach could democratize access to powerful AI technologies by reducing reliance on scarce, high-quality labeled data and enabling more inclusive participation in AI development. Furthermore, our method encourages the creation of robust AI systems capable of tackling complex problems, which can drive advancements in fields such as healthcare, education, and scientific research. However, careful consideration must be given to ethical implications, ensuring that the deployment of these advanced models aligns with societal values and mitigates risks associated with misuse or unintended consequences.

