# OpenReview forum: "EnsemW2S: Can an Ensemble of LLMs be Leveraged to Obtain a Stronger LLM?"
_NeurIPS.cc/2024/Workshop/SafeGenAi — SafeGenAi Poster_

### Official Review · Reviewer_evB8 · 2024-10-09
**Analysis of EnsemW2S-AdaBoost for Weak-to-Strong Generalization**

**Rating:** 7
**Confidence:** 2

**Review:**

This paper presents a new method for weak-to-strong generalization in LLMs that use weak models trained on simpler tasks to collectively supervise stronger models on more complex tasks. The authors introduce an AdaBoost-inspired ensemble algorithm to improve the performance of stronger LLMs on QA datasets.

Strength:
1. The paper is clearly written and well-motivated.
2. The proposed EnsemW2S-AdaBoost method is effective for using multiple weak LLMs for improving performance on complex tasks. This also shows promise in addressing the challenges of applying AdaBoost to generative AI models.
3. The experiments are comprehensive. It shows the effectiveness of ensemble learning in w2s generalization and hence will be of interest for the community.

Weaknesses:
1. The study mainly focuses on GPT-2 variants, which may not fully represent the diversity of LLM architectures available.
2. The experiment tasks are all multiple-choice on QA datasets. It would be interesting to see the performance on a broader range of NLP tasks.

---

### Official Review · Reviewer_HL5H · 2024-10-09
**Review of "ENSEMW2S: Can an Ensemble of LLMs Be Leveraged to Obtain a Stronger LLM?"**

**Rating:** 7
**Confidence:** 4

**Review:**

The paper tackles an important challenge in AI alignment: how to improve stronger LLMs using supervision from weaker models, especially in scenarios where high-quality labeled data is scarce.
## Quality
* Methodology: The proposed ensemble method is well-founded, drawing inspiration from AdaBoost, and is adapted thoughtfully for both classification and generative tasks.
* Experiments: The authors conduct experiments on binary classification and supervised fine-tuning tasks using multiple QA datasets, showing improvements over baselines.
* Results: They report up to 14% improvement over existing baselines and average improvements in both binary classification and generative tasks.
## Clarity
* Presentation: The paper is generally well-organized, with clear explanations of the proposed methods.
* Figures and Tables: Visual aids like figures and tables are used effectively to illustrate experimental results.
* Writing: The writing is technical but accessible, making the concepts understandable to readers familiar with LLMs and ensemble methods.
## Originality
* Novel Approach: Introducing an AdaBoost-inspired ensemble method to combine weak LLMs for w2s generalization is a novel contribution.
* Extension to Generative Tasks: Adapting AdaBoost for complex generation tasks and proposing the EnsemW2S-AdaBoost algorithm demonstrates originality.
## Significance
* Practical Impact: The work addresses the practical challenge of enhancing LLMs when high-quality supervision is limited, which is relevant for advancing AI alignment.
* Foundation for Future Research: It opens avenues for further exploration into ensemble methods for LLMs and the potential of weak models in supervising stronger ones.
## Pros
* Innovative Method: The ensemble approach effectively leverages multiple weak models to improve stronger LLMs.
* Comprehensive Experiments: The experiments cover both classification and generative tasks, providing a robust evaluation of the method.
* Addressing Real-World Challenges: The easy-to-hard framework reflects practical scenarios where simpler tasks are well-labeled, but complex tasks lack sufficient supervision.
## Cons
* Limited Task Diversity: The experiments are conducted on specific QA datasets; broader evaluation on diverse tasks could strengthen the findings.
* Detailed Analysis Missing: The paper could provide deeper insights into why the ensemble method outperforms single weak supervisors and the limitations encountered.